# THE RECURRENT NEURAL TANGENT KERNEL

**Sina Alemohammad, Zichao Wang, Randall Balestriero, Richard G. Baraniuk**
Department of Electrical and Computer Engineering
Rice University
`{sa86,zw16,rb42,richb}@rice.edu`

## ABSTRACT

The study of deep neural networks (DNNs) in the infinite-width limit, via the so-called *neural tangent kernel* (NTK) approach, has provided new insights into the dynamics of learning, generalization, and the impact of initialization. One key DNN architecture remains to be kernelized, namely, the recurrent neural network (RNN). In this paper we introduce and study the *Recurrent Neural Tangent Kernel* (RNTK), which provides new insights into the behavior of overparametrized RNNs. A key property of the RNTK should greatly benefit practitioners is its ability to compare inputs of different length. To this end, we characterize how the RNTK weights different time steps to form its output under different initialization parameters and nonlinearity choices. A synthetic and 56 real-world data experiments demonstrate that the RNTK offers significant performance gains over other kernels, including standard NTKs, across a wide array of data sets.

## 1 INTRODUCTION

The overparameterization of modern deep neural networks (DNNs) has resulted in not only remarkably good generalization performance on unseen data (Novak et al., 2018; Neyshabur et al., 2019; Belkin et al., 2019) but also guarantees that gradient descent learning can find the global minimum of their highly nonconvex loss functions (Du et al., 2019b; Allen-Zhu et al., 2019b;a; Zou et al., 2018; Arora et al., 2019b). From these successes, a natural question arises: What happens when we take overparameterization to the limit by allowing the width of a DNN's hidden layers to go to infinity? Surprisingly, the analysis of such an (impractical) DNN becomes analytically tractable. Indeed, recent work has shown that the training dynamics of (infinite-width) DNNs under gradient flow is captured by a constant kernel called the *Neural Tangent Kernel* (NTK) that evolves according to a linear ordinary differential equation (ODE) (Jacot et al., 2018; Lee et al., 2019; Arora et al., 2019a).

Every DNN architecture and parameter initialization produces a distinct NTK. The original NTK was derived from the Multilayer Perceptron (MLP)(Jacot et al., 2018) and was soon followed by kernels derived from Convolutional Neural Networks (CNTK) (Arora et al., 2019a; Yang, 2019a), Residual DNNs (Huang et al., 2020), and Graph Convolutional Neural Networks (GNTK) (Du et al., 2019a). In (Yang, 2020a), a general strategy to obtain the NTK of any architecture is provided.

***In this paper, we extend the NTK concept to the important class of overparametrized Recurrent Neural Networks (RNNs)***, a fundamental DNN architecture for processing sequential data. We show that RNN in its infinite-width limit converges to a kernel that we dub the ***Recurrent Neural Tangent Kernel*** (RNTK). The RNTK provides high performance for various machine learning tasks, and an analysis of the properties of the kernel provides useful insights into the behavior of RNNs in the following overparametrized regime. In particular, we derive and study the RNTK to answer the following theoretical questions:

***Q: Can the RNTK extract long-term dependencies between two data sequences?*** RNNs are known to underperform at learning long-term dependencies due to the gradient vanishing or exploding (Bengio et al., 1994). Attempted ameliorations have included orthogonal weights (Arjovsky et al., 2016; Jing et al., 2017; Henaff et al., 2016) and gating such as in Long Short-Term Memory (LSTM) (Hochreiter & Schmidhuber, 1997) and Gated Recurrent Unit (GRU) (Cho et al., 2014) RNNs. We demonstrate that the RNTK can detect long-term dependencies with proper initialization of the hyperparameters, and moreover, we show how the dependencies are extracted through time via different hyperparameter choices.

***Q: Do the recurrent weights of the RNTK reduce its representation power compared to other NTKs?*** An attractive property of an RNN that is shared by the RNTK is that it can deal with sequences of different lengths via weight sharing through time. This enables the reduction of the number of learnable parameters and thus more stable training at the cost of reduced representation power. We prove the surprising fact that employing tied vs. untied weights in an RNN *does not* impact the analytical form of the RNTK.

***Q: Does the RNTK generalize well?*** A recent study has revealed that the use of an SVM classifier with the NTK, CNTK, and GNTK kernels outperforms other classical kernel-based classifiers and trained finite DNNs on small data sets (typically fewer than 5000 training samples) (Lee et al., 2020; Arora et al., 2019a; 2020; Du et al., 2019a). We extend these results to RNTKs to demonstrate that the RNTK outperforms a variety of classic kernels, NTKs *and* finite RNNs for time series data sets in both classification and regression tasks. Carefully designed experiments with data of varying lengths demonstrate that the RNTK's performance accelerates beyond other techniques as the difference in lengths increases. Those results extend the empirical observations from (Arora et al., 2019a; 2020; Du et al., 2019a; Lee et al., 2020) into finite DNNs, NTK, CNTK, and GNTK comparisons by observing that their performance-wise ranking depends on the employed DNN architecture.

We summarize our contributions as follows:

**[C1]** We derive the analytical form for the RNTK of an overparametrized RNN at initialization using rectified linear unit (ReLU) and error function (erf) nonlinearities for arbitrary data lengths and number of layers (Section 3.1).

**[C2]** We prove that the RNTK remains constant during (overparametrized) RNN training and that the dynamics of training are simplified to a set of ordinary differential equations (ODEs) (Section 3.2).

**[C3]** When the input data sequences are of equal length, we show that the RNTKs of weight-tied and weight-untied RNNs converge to the same RNTK (Section 3.3).

**[C4]** Leveraging our analytical formulation of the RNTK, we empirically demonstrate how correlations between data at different times are weighted by the function learned by an RNN for different sets of hyperparameters. We also offer practical suggestions for choosing the RNN hyperparameters for deep information propagation through time (Section 3.4).

**[C5]** We demonstrate that the RNTK is eminently practical by showing its superiority over classical kernels, NTKs, and finite RNNs in exhaustive experiments on time-series classification and regression with both synthetic and 56 real-world data sets (Section 4).

## 2 BACKGROUND AND RELATED WORK

**Notation.** We denote $[n] = \{1, \ldots, n\}$, and $\boldsymbol{I}_d$ the identity matrix of size $d$. $[\boldsymbol{A}]_{i,j}$ represents the $(i, j)$-th entry of a matrix, and similarly $[\boldsymbol{a}]_i$ represents the $i$-th entry of a vector. We use $\phi(\cdot) : \mathbb{R} \to \mathbb{R}$ to represent the activation function that acts coordinate wise on a vector and $\phi'$ to denote its derivative. We will often use the rectified linear unit (ReLU) $\phi(x) = \max(0, x)$ and error function (erf) $\phi(x) = \frac{2}{\sqrt{\pi}} \int_0^x e^{-z^2} dz$ activation functions. $\mathcal{N}(\boldsymbol{\mu}, \boldsymbol{\Sigma})$ represents the multidimensional Gaussian distribution with the mean vector $\boldsymbol{\mu}$ and the covariance matrix $\boldsymbol{\Sigma}$.

**Recurrent Neural Networks (RNNs).** Given an input sequence data $\boldsymbol{x} = \{\boldsymbol{x}_t\}_{t=1}^T$ of length $T$ with data at time $t$, $\boldsymbol{x}_t \in \mathbb{R}^m$, a *simple RNN* (Elman, 1990) performs the following recursive computation at each layer $\ell$ and each time step $t$

$$\boldsymbol{g}^{(\ell,t)}(\boldsymbol{x}) = \boldsymbol{W}^{(\ell)}\boldsymbol{h}^{(\ell,t-1)}(\boldsymbol{x}) + \boldsymbol{U}^{(\ell)}\boldsymbol{h}^{(\ell-1,t)}(\boldsymbol{x}) + \boldsymbol{b}^{(\ell)}, \qquad \boldsymbol{h}^{(\ell,t)}(\boldsymbol{x}) = \phi\left(\boldsymbol{g}^{(\ell,t)}(\boldsymbol{x})\right),$$

where $\boldsymbol{W}^{(\ell)} \in \mathbb{R}^{n \times n}$, $\boldsymbol{b}^{(\ell)} \in \mathbb{R}^n$ for $\ell \in [L]$, $\boldsymbol{U}^{(1)} \in \mathbb{R}^{n \times m}$ and $\boldsymbol{U}^{(\ell)} \in \mathbb{R}^{n \times n}$ for $\ell \geq 2$ are the RNN parameters. $\boldsymbol{g}^{(\ell,t)}(\boldsymbol{x})$ is the pre-activation vector at layer $\ell$ and time step $t$, and $\boldsymbol{h}^{(\ell,t)}(\boldsymbol{x})$ is the after-activation (hidden state). For the input layer $\ell = 0$, we define $\boldsymbol{h}^{(0,t)}(\boldsymbol{x}) := \boldsymbol{x}_t$. $\boldsymbol{h}^{(\ell,0)}(\boldsymbol{x})$ as the initial hidden state at layer $\ell$ that must be initialized to start the RNN recursive computation.

The output of an $L$-hidden layer RNN with linear read out layer is achieved via

$$f_\theta(\boldsymbol{x}) = \boldsymbol{V}\boldsymbol{h}^{(L,T)}(\boldsymbol{x}),$$

where $\boldsymbol{V} \in \mathbb{R}^{d \times n}$. Figure 1 visualizes an RNN unrolled through time.

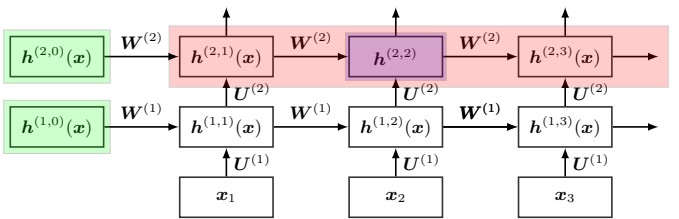

Figure 1: Visualization of a simple RNN that highlights a cell (purple), a layer (red) and the initial hidden state of each layer (green). (Best viewed in color.)

**Neural Tangent Kernel (NTK).** Let $f_\theta(\boldsymbol{x}) \in \mathbb{R}^d$ be the output of a DNN with parameters $\theta$. For two input data sequences $\boldsymbol{x}$ and $\boldsymbol{x}'$, the NTK is defined as (Jacot et al., 2018)

$$\widehat{\Theta}_s(\boldsymbol{x}, \boldsymbol{x}') = \langle \nabla_{\theta_s} f_{\theta_s}(\boldsymbol{x}), \nabla_{\theta_s} f_{\theta_s}(\boldsymbol{x}') \rangle,$$

where $f_{\theta_s}$ and $\theta_s$ are the network output and parameters during training at time s.[1] Let $\mathcal{X}$ and $\mathcal{Y}$ be the set of training inputs and targets, $\ell(\widehat{y}, y) : \mathbb{R}^d \times \mathbb{R}^d \to \mathbb{R}^+$ be the loss function, and $\mathcal{L} = \frac{1}{|\mathcal{X}|} \sum_{(\boldsymbol{x}, \boldsymbol{y}) \in \mathcal{X} \times \mathcal{Y}} \ell(f_{\theta_s}(\boldsymbol{x}), \boldsymbol{y})$ be the the empirical loss. The evolution of the parameters $\theta_s$ and output of the network $f_{\theta_s}$ on a test input using gradient descent with infinitesimal step size (a.k.a gradient flow) with learning rate $\eta$ is given by

$$\frac{\partial \theta_s}{\partial s} = -\eta \nabla_{\theta_s} f_{\theta_s}(\mathcal{X})^T \nabla_{f_{\theta_s}(\mathcal{X})} \mathcal{L} \tag{1}$$

$$\frac{\partial f_{\theta_s}(\boldsymbol{x})}{\partial s} = -\eta \nabla_{\theta_s} f_{\theta_s}(\boldsymbol{x}) \nabla_{\theta_s} f_{\theta_s}(\mathcal{X})^T \nabla_{f_{\theta_s}(\mathcal{X})} \mathcal{L} = -\eta \widehat{\Theta}_s(\boldsymbol{x}, \mathcal{X}) \nabla_{f_{\theta_s}(\mathcal{X})} \mathcal{L}. \tag{2}$$

Generally, $\widehat{\Theta}_s(\boldsymbol{x}, \boldsymbol{x}')$, hereafter referred to as the empirical NTK, changes over time during training, making the analysis of the training dynamics difficult. When $f_{\theta_s}$ corresponds to an infinite-width MLP, (Jacot et al., 2018) showed that $\widehat{\Theta}_s(\boldsymbol{x}, \boldsymbol{x}')$ converges to a limiting kernel at initialization and stays constant during training, i.e.,

$$\lim_{n \to \infty} \widehat{\Theta}_s(\boldsymbol{x}, \boldsymbol{x}') = \lim_{n \to \infty} \widehat{\Theta}_0(\boldsymbol{x}, \boldsymbol{x}') := \Theta(\boldsymbol{x}, \boldsymbol{x}') \quad \forall s,$$

which is equivalent to replacing the outputs of the DNN by their first-order Taylor expansion in the parameter space (Lee et al., 2019). With a mean-square error (MSE) loss function, the training dynamics in (1) and (2) simplify to a set of linear ODEs, which coincides with the training dynamics of kernel ridge regression with respect to the NTK when the ridge term goes to zero. A nonzero ridge regularization can be conjured up by adding a regularization term $\frac{\lambda^2}{2} \|\theta_s - \theta_0\|_2^2$ to the empirical loss (Hu et al., 2020).

## 3    THE RECURRENT NEURAL TANGENT KERNEL

We are now ready to derive the RNTK. We first prove the convergence of an RNN at initialization to the RNTK in the infinite-width limit and discuss various insights it provides. We then derive the convergence of an RNN after training to the RNTK. Finally, we analyze the effects of various hyperparameter choices on the RNTK. Proofs of all of our results are provided in the Appendices.

### 3.1    RNTK FOR AN INFINITE-WIDTH RNN AT INITIALIZATION

First we specify the following parameter initialization scheme that follows previous work on NTKs (Jacot et al., 2018), which is crucial to our convergence results:

$$\boldsymbol{W}^{(\ell)} = \frac{\sigma_w^\ell}{\sqrt{n}} \mathbf{W}^{(\ell)}, \quad \boldsymbol{U}^{(1)} = \frac{\sigma_u^1}{\sqrt{m}} \mathbf{U}^{(1)}, \quad \boldsymbol{U}^{(\ell)} = \frac{\sigma_u^\ell}{\sqrt{n}} \mathbf{U}^{(\ell)} (\ell \geq 2), \quad \boldsymbol{V} = \frac{\sigma_v}{\sqrt{n}} \mathbf{V}, \quad \boldsymbol{b}^{(\ell)} = \sigma_b \mathbf{b}^{(\ell)}, \tag{3}$$

where

$$[\mathbf{W}^\ell]_{i,j}, \ [\mathbf{U}^{(\ell)}]_{i,j}, \ [\mathbf{V}]_{i,j}, \ [\mathbf{b}^{(\ell)}]_i \sim \mathcal{N}(0, 1). \tag{4}$$

We will refer to (3) and (4) as the *NTK initialization*. The choices of the hyperparameters $\sigma_w$, $\sigma_u$, $\sigma_v$ and $\sigma_b$ can significantly impact RNN performance, and we discuss them in detail in Section

---

[1]We use $s$ to denote time here, since $t$ is used to index the time steps of the RNN inputs.

3.4. For the initial (at time $t = 0$) hidden state at each layer $\ell$, we set $\boldsymbol{h}^{(\ell,0)}(\boldsymbol{x})$ to an i.i.d. copy of $\mathcal{N}(0, \sigma_h)$ (Wang et al., 2018) . For convenience, we collect all of the learnable parameters of the RNN into $\theta = \text{vect}\big[\{\{\mathbf{W}^{(\ell)}, \mathbf{U}^{(\ell)}, \mathbf{b}^{(\ell)}\}_{\ell=1}^{L}, \mathbf{V}\}\big]$.

The derivation of the RNTK at initialization is based on the correspondence between Gaussian initialized, infinite-width DNNs and Gaussian Processes (GPs), known as the DNN-GP. In this setting every coordinate of the DNN output tends to a GP as the number of units/neurons in the hidden layer (its width) goes to infinity. The corresponding DNN-GP kernel is computed as

$$\mathcal{K}(\boldsymbol{x}, \boldsymbol{x}') = \underset{\theta \sim \mathcal{N}}{\mathbb{E}}\big[[f_\theta(\boldsymbol{x})]_i \cdot [f_\theta(\boldsymbol{x}')]_i\big], \ \forall i \in [d]. \tag{5}$$

First introduced for a single-layer, fully-connected neural network by (Neal, 1995), recent works on NTKs have extended the results for various DNN architectures (Lee et al., 2018; Duvenaud et al., 2014; Novak et al., 2019; Garriga-Alonso et al., 2019; Yang, 2019b), where in addition to the output, all pre-activation layers of the DNN tends to a GPs in the infinite-width limit. In the case of RNNs, each coordinate of the RNN pre-activation $\boldsymbol{g}^{(\ell,t)}(\boldsymbol{x})$ converges to a centered GP depending on the inputs with kernel

$$\Sigma^{(\ell,t,t')}(\boldsymbol{x}, \boldsymbol{x}') = \underset{\theta \sim \mathcal{N}}{\mathbb{E}}\big[[\boldsymbol{g}^{(\ell,t)}(\boldsymbol{x})]_i \cdot [\boldsymbol{g}^{(\ell,t')}(\boldsymbol{x}')]_i\big] \ \forall i \in [n]. \tag{6}$$

As per (Yang, 2019a), the gradients of random infinite-width DNNs computed during backpropagation are also Gaussian distributed. In the case of RNNs, every coordinate of the vector $\boldsymbol{\delta}^{(\ell,t)}(\boldsymbol{x}) := \sqrt{n}\big(\nabla_{\boldsymbol{g}^{(\ell,t)}(\boldsymbol{x})} f_\theta(\boldsymbol{x})\big)$ converges to a GP with kernel

$$\Pi^{(\ell,t,t')}(\boldsymbol{x}, \boldsymbol{x}') = \underset{\theta \sim \mathcal{N}}{\mathbb{E}}\big[[\boldsymbol{\delta}^{(\ell,t)}(\boldsymbol{x})]_i \cdot [\boldsymbol{\delta}^{(\ell,t')}(\boldsymbol{x}')]_i\big] \ \forall i \in [n]. \tag{7}$$

Both convergences occur independently of the coordinate index $i$ and for inputs of possibly different lengths, i.e., $T \neq T'$. With (6) and (7), we now prove that an infinite-width RNN at initialization converges to the limiting RNTK.

**Theorem 1** *Let $\boldsymbol{x}$ and $\boldsymbol{x}'$ be two data sequences of potentially different lengths $T$ and $T'$, respectively. Without loss of generality, assume that $T \leq T'$, and let $\tau := T' - T$. Let $n$ be the number of units in the hidden layers, the empirical RNTK for an L-layer RNN with NTK initialization converges to the following limiting kernel as $n \to \infty$*

$$\lim_{n \to \infty} \widehat{\Theta}_0(\boldsymbol{x}, \boldsymbol{x}') = \Theta(\boldsymbol{x}, \boldsymbol{x}') = \Theta^{(L,T,T')}(\boldsymbol{x}, \boldsymbol{x}') \otimes \boldsymbol{I}_d, \tag{8}$$

*where*

$$\Theta^{(L,T,T')}(\boldsymbol{x}, \boldsymbol{x}') = \left(\sum_{\ell=1}^{L} \sum_{t=1}^{T} \Big(\Pi^{(\ell,t,t+\tau)}(\boldsymbol{x}, \boldsymbol{x}') \cdot \Sigma^{(\ell,t,t+\tau)}(\boldsymbol{x}, \boldsymbol{x}')\Big)\right) + \mathcal{K}(\boldsymbol{x}, \boldsymbol{x}'), \tag{9}$$

*with $\mathcal{K}(\boldsymbol{x}, \boldsymbol{x}')$, $\Sigma^{(\ell,t,t+\tau)}(\boldsymbol{x}, \boldsymbol{x}')$, and $\Pi^{(\ell,t,t+\tau)}(\boldsymbol{x}, \boldsymbol{x}')$ defined in (5)–(7).*

**Remarks.** Theorem 1 holds generally for any two data sequences, including different lengths ones. This highlights the RNTK's ability to produce a similarity measure $\Theta(\boldsymbol{x}, \boldsymbol{x}')$ even if the inputs are of different lengths, without resorting to heuristics such as zero padding the inputs to the to the max length of both sequences. Dealing with data of different length is in sharp contrast to common kernels such as the classical radial basis functions, polynomial kernels, and current NTKs. We showcase this capability below in Section 4.

To visualize Theorem 1, we plot in the left plot in Figure 2 the convergence of a single layer, sufficiently wide RNN to its RNTK with the two simple inputs $\boldsymbol{x} = \{1, -1, 1\}$ of length 3 and $\boldsymbol{x}' = \{\cos(\alpha), \sin(\alpha)\}$ of length 2, where $\alpha = [0, 2\pi]$. For an RNN with a sufficiently large hidden state ($n = 1000$), we see clearly that it converges to the RNTK ($n = \infty$).

**RNTK Example for a Single-Layer RNN.** We present a concrete example of Theorem 1 by showing how to recursively compute the RNTK for a single-layer RNN; thus we drop the layer index for notational simplicity. ***We compute and display the RNTK for the general case of a multi-layer RNN in Appendix B.3***. To compute the RNTK $\Theta^{(T,T')}(\boldsymbol{x}, \boldsymbol{x}')$, we need to compute the GP

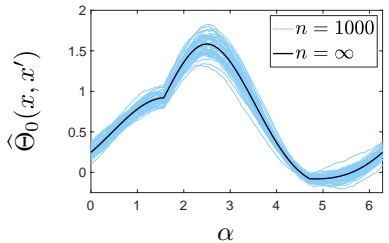 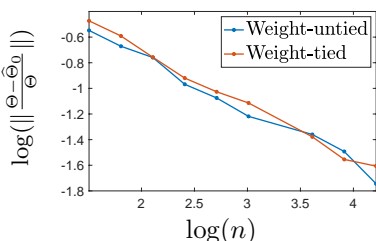

Figure 2: Empirical demonstration of a wide, single-layer RNN converging to its limiting RNTK. **Left**: convergence for a pair of different-length inputs $\boldsymbol{x} = \{1, -1, 1\}$ and $\boldsymbol{x}' = \{\cos(\alpha), \sin(\alpha)\}$, with varying $\alpha = [0, 2\pi]$. The vertical axis corresponds to the RNTK values for different values of $\alpha$. **Right**: convergence of weight-tied and weight-untied single layer RNN to the same limiting RNTK with increasing width (horizontal axis). The vertical axis corresponds to the average of the log-normalized error between the empirical RNTK computed using finite RNNs and the RNTK for 50 Gaussian normal signals of length $T = 5$.

kernels $\Sigma^{(t,t+\tau)}(\boldsymbol{x}, \boldsymbol{x}')$ and $\Pi^{(t,t+\tau)}(\boldsymbol{x}, \boldsymbol{x}')$. We first define the operator $\mathrm{V}_\phi\big[\boldsymbol{K}\big]$ that depends on the nonlinearity $\phi(\cdot)$ and a positive semi-definite matrix $\boldsymbol{K} \in \mathbb{R}^{2\times 2}$

$$\mathrm{V}_\phi\big[\boldsymbol{K}\big] = \mathbb{E}[\phi(\mathrm{z}_1) \cdot \phi(\mathrm{z}_2)], \qquad (\mathrm{z}_1, \mathrm{z}_2) \sim \mathcal{N}(0, \boldsymbol{K}). \tag{10}$$

Following (Yang, 2019a), we obtain the analytical recursive formula for the GP kernel $\Sigma^{(t,t+\tau)}(\boldsymbol{x}, \boldsymbol{x}')$ for a single layer RNN as

$$\Sigma^{(1,1)}(\boldsymbol{x}, \boldsymbol{x}') = \sigma_w^2 \sigma_h^2 1_{(\boldsymbol{x}=\boldsymbol{x}')} + \frac{\sigma_u^2}{m}\langle\boldsymbol{x}_1, \boldsymbol{x}_1'\rangle + \sigma_b^2 \tag{11}$$

$$\Sigma^{(t,1)}(\boldsymbol{x}, \boldsymbol{x}') = \frac{\sigma_u^2}{m}\langle\boldsymbol{x}_t, \boldsymbol{x}_1'\rangle + \sigma_b^2 \qquad\qquad t > 1 \tag{12}$$

$$\Sigma^{(1,t')}(\boldsymbol{x}, \boldsymbol{x}') = \frac{\sigma_u^2}{m}\langle\boldsymbol{x}_1, \boldsymbol{x}_{t'}'\rangle + \sigma_b^2 \qquad\qquad t' > 1 \tag{13}$$

$$\Sigma^{(t,t')}(\boldsymbol{x}, \boldsymbol{x}') = \sigma_w^2 \mathrm{V}_\phi\big[\boldsymbol{K}^{(t,t')}(\boldsymbol{x}, \boldsymbol{x}')\big] + \frac{\sigma_u^2}{m}\langle\boldsymbol{x}_t, \boldsymbol{x}_{t'}'\rangle + \sigma_b^2 \qquad t, t' > 1 \tag{14}$$

$$\mathcal{K}(\boldsymbol{x}, \boldsymbol{x}') = \sigma_v^2 \mathrm{V}_\phi\big[\boldsymbol{K}^{(T+1,T'+1)}(\boldsymbol{x}, \boldsymbol{x}')\big], \tag{15}$$

where

$$\boldsymbol{K}^{(t,t')}(\boldsymbol{x}, \boldsymbol{x}') = \left[ \begin{array}{cc} \Sigma^{(t-1,t-1)}(\boldsymbol{x}, \boldsymbol{x}) & \Sigma^{(t-1,t'-1)}(\boldsymbol{x}, \boldsymbol{x}') \\ \Sigma^{(t-1,t'-1)}(\boldsymbol{x}, \boldsymbol{x}') & \Sigma^{(t'-1,t'-1)}(\boldsymbol{x}', \boldsymbol{x}') \end{array} \right]. \tag{16}$$

Similarly, we obtain the analytical recursive formula for the GP kernel $\Pi^{(t,t+\tau)}(\boldsymbol{x}, \boldsymbol{x}')$ as

$$\Pi^{(T,T')}(\boldsymbol{x}, \boldsymbol{x}') = \sigma_v^2 \mathrm{V}_{\phi'}\big[\boldsymbol{K}^{(T+1,T+\tau+1)}(\boldsymbol{x}, \boldsymbol{x}')\big] \tag{17}$$

$$\Pi^{(t,t+\tau)}(\boldsymbol{x}, \boldsymbol{x}') = \sigma_w^2 \mathrm{V}_{\phi'}\big[\boldsymbol{K}^{(t+1,t+\tau+1)}(\boldsymbol{x}, \boldsymbol{x}')\big]\Pi^{(t+1,t+1+\tau)}(\boldsymbol{x}, \boldsymbol{x}') \qquad t \in [T-1] \tag{18}$$

$$\Pi^{(t,t')}(\boldsymbol{x}, \boldsymbol{x}') = 0 \qquad\qquad t' - t \neq \tau. \tag{19}$$

For $\phi = \mathrm{ReLU}$ and $\phi = \mathrm{erf}$, we provide analytical expressions for $\mathrm{V}_\phi\big[\boldsymbol{K}\big]$ and $\mathrm{V}_{\phi'}\big[\boldsymbol{K}\big]$ in Appendix B.5. These yield an explicit formula for the RNTK that enables fast and point-wise kernel evaluations. For other activation functions, one can apply the Monte Carlo approximation to obtain $\mathrm{V}_\phi\big[\boldsymbol{K}\big]$ and $\mathrm{V}_{\phi'}\big[\boldsymbol{K}\big]$ (Novak et al., 2019).

### 3.2 RNTK for an Infinite-Width RNN during Training

We prove that an infinitely-wide RNN, not only at initialization but also *during* gradient descent training, converges to the limiting RNTK at initialization.

**Theorem 2** *Let $n$ be the number of units of each RNN's layer. Assume that $\Theta(\mathcal{X}, \mathcal{X})$ is positive definite on $\mathcal{X}$ such that $\lambda_{\min}(\Theta(\mathcal{X}, \mathcal{X})) > 0$. Let $\eta^* := 2\big(\lambda_{\min}(\Theta(\mathcal{X}, \mathcal{X})) + \lambda_{\max}(\Theta(\mathcal{X}, \mathcal{X}))\big)^{-1}$. For an $L$-layer RNN with NTK initialization as in (3), (4) trained under gradient flow (recall (1) and (2)) with $\eta < \eta^*$, we have with high probability*

$$\sup_s \frac{\|\theta_s - \theta_0\|_2}{\sqrt{n}}, \sup_s \|\widehat{\Theta}_s(\mathcal{X}, \mathcal{X}) - \widehat{\Theta}_0(\mathcal{X}, \mathcal{X})\|_2 = \mathcal{O}\left(\frac{1}{\sqrt{n}}\right).$$

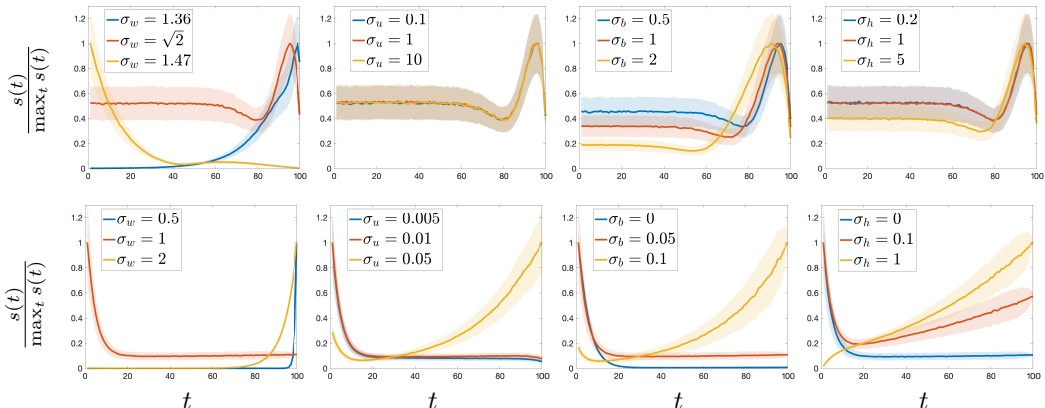

Figure 3: Per time step $t$ (horizontal axis) sensitivity analysis (vertical axis) of the RNTK for the ReLU (top row) and erf (bottom row) activation functions for various weight noise hyperparameters. We also experiment with different RNTK hyperparameters in each of the subplots, given by the subplot internal legend. Clearly, the ReLU (top-row) provides a more stable kernel across time steps (highlighted by the near constant sensitivity through time). On the other hand, erf (bottom row) sees a more erratic behavior either focusing entirely on early time-steps or on the latter ones.

**Remarks.** Theorem 2 states that the training dynamics of an RNN in the infinite-width limit as in (1), (2) are governed by the RNTK derived from the RNN at its initialization. Intuitively, this is due to the NTK initialization (3), (4) which positions the parameters near a local minima, thus minimizing the amount of update that needs to be applied to the weights to obtain the final parameters.

### 3.3 RNTK FOR AN INFINITE-WIDTH RNN WITHOUT WEIGHT SHARING

We prove that, in the infinite-width limit, an RNN without weight sharing (untied weights), i.e., using independent new weights $\mathbf{W}^{(\ell,t)}$, $\mathbf{U}^{(\ell,t)}$ and $\mathbf{b}^{(\ell,t)}$ at each time step $t$, converges to the same RNTK as an RNN with weight sharing (tied weights). First, recall that it is a common practice to use weight-tied RNNs, i.e., in layer $\ell$, the weights $\mathbf{W}^{(\ell)}$, $\mathbf{U}^{(\ell)}$ and $\mathbf{b}^{(\ell)}$ are the same across all time steps $t$. This practice conserves memory and reduces the number of learnable parameters. We demonstrate that, when using untied-weights, the RNTK formula remains unchanged.

**Theorem 3** *For inputs of the same length, an RNN with untied weights converges to the same RNTK as an RNN with tied weights in the infinite-width ($n \to \infty$) regime.*

**Remarks.** Theorem 3 implies that weight-tied and weight-untied RNNs have similar behaviors in the infinite-width limit. It also suggests that existing results on the simpler, weight-untied RNN setting may be applicable for the more general, weight-tied RNN. The plot on the right side of Figure 2 empirically demonstrates the convergence of both the weight-tied and weight-untied RNNs to the RNTK with increasing hidden layer size $n$; moreover, the convergence rates are similar.

### 3.4 INSIGHTS INTO THE ROLES OF THE RNTK'S HYPERPARAMETERS

Our analytical form for the RNTK is fully determined by a small number of hyperparameters, which contains the various weight variances collected into $\mathcal{S} = \{\sigma_w, \sigma_u, \sigma_b, \sigma_h\}$ and the activation function.[2] In standard supervised-learning settings, one often performs cross-validation to select the hyperparameters. However, since kernel methods become computationally intractable for large datasets, we seek a more computationally friendly alternative to cross-validation. Here we conduct a novel exploratory analysis that provides new insights into the impact of the RNTK hyperparameters on the RNTK output and suggests a simple method to select them a priori in a deliberate manner.

To visualize the role of the RNTK hyperparameters, we introduce the *sensitivity* $s(t)$ of the RNTK of two input sequences $\boldsymbol{x}$ and $\boldsymbol{x}'$ with respect to the input $\boldsymbol{x}_t$ at time $t$

$$s(t) = \|\nabla_{\boldsymbol{x}_t} \Theta(\boldsymbol{x}, \boldsymbol{x}')\|_2. \tag{20}$$

---

[2]From (11) to (18) we emphasize that $\sigma_v$ merely scales the RNTK and does not change its overall behavior.

Table 1: Summary of time series classification results on 56 real-world data sets. The RNTK outperforms classical kernels, the NTK, and trained RNNs across all metrics. See Appendix A for detailed description of the metrics.

| | RNTK | NTK | RBF | Polynomial | Gaussian RNN | Identity RNN | GRU |
|---|---|---|---|---|---|---|---|
| Acc. mean ↑ | **80.15% ± 15.99%** | 77.74% ± 16.61% | 78.15% ± 16.59% | 77.69% ± 16.40% | 55.98% ± 26.42% | 63.08% ± 19.02 % | 69.50% ± 22.67 |
| P90 ↑ | **92.86%** | 85.71% | 87.60% | 82.14% | 28.57% | 42.86% | 60.71% |
| P95 ↑ | **80.36%** | 66.07% | 75.00% | 67.86% | 17.86% | 21.43% | 46.43% |
| PMA ↑ | **97.23%** | 94.30% | 94.86% | 94.23% | 67.06% | 78.22% | 84.31% |
| Friedman Rank ↓ | **2.38** | 3.00 | 2.89 | 3.46 | 5.86 | 5.21 | 4.21 |

Here, $s(t)$ indicates how sensitive the RNTK is to the data at time $t$, i.e., $\boldsymbol{x}_t$, in presence of another data sequence $\boldsymbol{x}'$. Intuitively, large/small $s(t)$ indicates that the RNTK is relatively sensitive/insensitive to the input $\boldsymbol{x}_t$ at time $t$.

The sensitivity is crucial to understanding to which extent the RNTK prediction is impacted by the input at each time step. In the case where some time indices have a small sensitivity, then any input variation in those corresponding times will not alter the RNTK output and thus will produce a metric that is invariant to those changes. This situation can be beneficial or detrimental based on the task at hand. Ideally, and in the absence of prior knowledge on the data, one should aim to have a roughly constant sensitivity across time in order to treat all time steps equally in the RNTK input comparison.

Figure 3 plots the normalized sensitivity $s(t)/\max_t(s(t))$ for two data sequences of the same length $T = 100$, with $s(t)$ computed numerically for $\boldsymbol{x}_t, \boldsymbol{x}'_t \sim \mathcal{N}(0, 1)$. We repeated the experiments 10000 times; the mean of the sensitivity is shown in Figure 3. Each of the plots shows the changes of parameters $\mathcal{S}_{\mathrm{ReLU}} = \{\sqrt{2}, 1, 0, 0\}$ for $\phi = \mathrm{ReLU}$ and $\mathcal{S}_{\mathrm{erf}} = \{1, 0.01, 0.05, 0\}$ for $\phi = \mathrm{erf}$.

From Figure 3 we first observe that both $\mathrm{ReLU}$ and $\mathrm{erf}$ show similar per time step sensitivity measure $s(t)$ behavior around the hyperparameters $\mathcal{S}_{\mathrm{ReLU}}$ and $\mathcal{S}_{\mathrm{erf}}$. If one varies any of the weight variance parameters, the sensitivity exhibits a wide range of behavior, and in particular with $\mathrm{erf}$. We observe that $\sigma_w$ has a major influence on $s(t)$. For ReLU, a small decrease/increase in $\sigma_w$ can lead to over-sensitivity of the RNTK to data at the last/first times steps, whereas for $\mathrm{erf}$, any changes in $\sigma_w$ leads to over-sensitivity to the last time steps.

Another notable observation is the importance of $\sigma_h$, which is usually set to zero for RNNs. (Wang et al., 2018) showed that a non-zero $\sigma_h$ acts as a regularization that improves the performance of RNNs with the $\mathrm{ReLU}$ nonlinearity. From the sensitivity perspective, a non-zero $\sigma_h$ results in reducing the importance of the first time steps of the input. We also see the same behavior in $\mathrm{erf}$, but with stronger changes as $\sigma_h$ increases. Hence whenever one aims at reinforcing the input pairwise comparisons, such parameters should be favored.

This sensitivity analysis provides a practical tool for RNTK hyperparameter tuning. In the absence of knowledge about the data, hyperparameters should be chosen to produce the least time varying sensitivity. If given a priori knowledge, hyperparameters can be selected that direct the RNTK to the desired time-steps.

## 4 EXPERIMENTS

We now empirically validate the performance of the RNTK compared to classic kernels, NTKs, and trained RNNs on both classification and regression tasks using a large number of time series data sets. Of particular interest is the capability of the RNTK to offer high performance even on inputs of different lengths.

**Time Series Classification.** The first set of experiments considers time series inputs of the same lengths from 56 datasets in the UCR time-series classification data repository (Dau et al., 2019). We restrict ourselves to selected data sets with fewer than 1000 training samples and fewer than 1000 time steps ($T$) as kernel methods become rapidly intractable for larger datasets. We compare the RNTK with a variety of other kernels, including the Radial Basis Kernel (RBF), polynomial kernel, and NTK (Jacot et al., 2018), as well as finite RNNs with Gaussian, identity (Le et al., 2015) initialization, and GRU (Cho et al., 2014). We use $\phi = \mathrm{ReLU}$ for both the RNTKs and NTKs. For each kernel, we train a C-SVM (Chang & Lin, 2011) classifier, and for each finite RNN we use gradient descent training. For model hyperparameter tuning, we use 10-fold cross-validation. Details on the data sets and experimental setup are available in Appendix A.1.

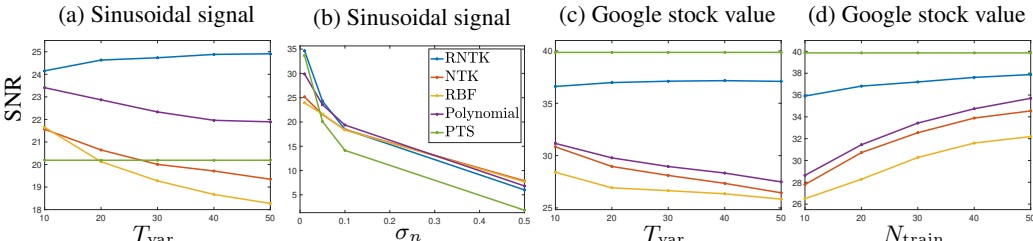

Figure 4: Performance of the RNTK on the synthetic sinusoid and real-world Google stock price data sets compared to three other kernels. We vary the input lengths (a,c), the input noise level (b), and training set size (d). We compute the average SNR by repeating each experiment 1000 times. The RNTK clearly outperforms all of the other kernels under consideration. Figure 4b suggests that the RNTK performs better when input noise level is low demonstrating one case where time recurrence from RNTK might be sub-optimal as it collects and accumulate the high noise from each time step as opposed to other kernels treating each independently.

We summarize the classification results over all 56 datasets in Table 1; detailed results on each data set is available in Appendix A.2. We see that the RNTK outperforms not only the classical kernels but also the NTK and trained RNNs in all metrics. The results demonstrate the ability of RNTK to provide increased performances compare to various other methods (kernels and RNNs). The superior performance of RNTK compared to other kernels, including NTK, can be explained by the internal recurrent mechanism present in RNTK, allowing time-series adapted sample comparison. In addition, RNTK also outperforms RNN and GRU. As the datasets we consider are relative small in size, finite RNNs and GRUs that typically require large amount of data to succeed do not perform well in our setting. An interesting future direction would be to compare RNTK to RNN/GRU on larger datasets.

**Time Series Regression.** We now validate the performance of the RNTK on time series inputs of *different* lengths on both synthetic data and real data. For both scenarios, the target is to predict the next time-step observation of the randomly extracted windows of different length using kernel ridge regression.

We compare the RNTK to other kernels, the RBF and polynomial kernels and the NTK. We also compare our results with a data independent predictor that requires no training, that is simply to predict the next time step with previous time step (PTS).

For the synthetic data experiment, we simulate 1000 samples of one period of a sinusoid and add white Gaussian noise with default $\sigma_n = 0.05$. From this fixed data, we extract training set size $N_{\text{train}} = 20$ segments of uniform random lengths in the range of $[T_{\text{fixed}}, T_{\text{fixed}} + T_{\text{var}}]$ with $T_{\text{fixed}} = 10$. We use standard kernel ridge regression for this task. The test set is comprised of $N_{\text{test}} = 5000$ obtained from other randomly extracted segments, again of varying lengths. For the real data, we use 975 days of the Google stock value in the years 2014–2018. As in the simulated signal setup above, we extract $N_{\text{train}}$ segments of different lengths from the first 700 days and test on the $N_{\text{test}}$ segments from days 701 to 975. Details of the experiment are available in Appendix A.2.

We report the predicted signal-to-noise ratio (SNR) for both datasets in Figures 4a and 4c for various values of $T_{\text{var}}$. We vary the noise level and training set size for fixed $T_{\text{var}} = 10$ in Figures 4b and 4d. As we see from Figures 4a and 4c, the RNTK offers substantial performance gains compared to the other kernels, due to its ability to naturally deal with variable length inputs. Moreover, the performance gap increases with the amount of length variation of the inputs $T_{\text{var}}$. Figure 4d demonstrates that, unlike the other methods, the RNTK maintains its performance even when the training set is small. Finally, Figure 4c demonstrates that the impact of noise in the data on the regression performance is roughly the same for all models but becomes more important for RNTK with a large $\sigma_n$; this might be attributed to the recurrent structure of the model allowing for a time propagation and amplification of the noise for very low SNR. These experiments demonstrate the distinctive advantages of the RNTK over classical kernels, and NTKs for input data sequences of varying lengths.

In the case of PTS, we expect the predictor to outperform kernel methods when learning from the training samples is hard, due to noise in the data or small training size which can lead to over fitting. In Figure 4a RNTK and Polynomial kernels outperforms PTS for all values of $T_{\text{var}}$, but for larger $T_{\text{var}}$, NTK and RBF under perform PTS due to the increasing detrimental effect of zero padding.

For the Google stock value, we see a superior performance of PTS with respect to all other kernel methods due to the nature of those data heavily relying on close past data. However, RNTK is able to reduce the effect of over-fitting, and provide the closest results to PTS among all kernel methods we employed, with increasing performance as the number of training samples increase.

## 5 CONCLUSIONS

In this paper, we have derived the RNTK based on the architecture of a simple RNN. We have proved that, at initialization, after training, and without weight sharing, any simple RNN converges to the same RNTK. This convergence provides new insights into the behavior of infinite-width RNNs, including how they process different-length inputs, their training dynamics, and the sensitivity of their output at every time step to different nonlinearities and initializations. We have highlighted the RNTK's practical utility by demonstrating its superior performance on time series classification and regression compared to a range of classical kernels, the NTK, and trained RNNs. There are many avenues for future research, including developing RNTKs for gated RNNs such as the LSTM (Hochreiter & Schmidhuber, 1997) and investigating which of our theoretical insights extend to finite RNNs.

## ACKNOWLEDGMENTS

This work was supported by NSF grants CCF-1911094, IIS-1838177, and IIS-1730574; ONR grants N00014-18-12571, N00014-20-1-2787, and N00014-20-1-2534; AFOSR grant FA9550-18-1-0478; and a Vannevar Bush Faculty Fellowship, ONR grant N00014-18-1-2047.

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

# A  EXPERIMENT DETAILS

## A.1  TIME SERIES CLASSIFICATION

**Kernel methods settings.**    We used RNTK, RBF, polynomial and NTK (Jacot et al., 2018). For data pre-processing, we normalized the norm of each $\boldsymbol{x}$ to 1. For training we used C-SVM in LIBSVM library (Chang & Lin, 2011) and for hyperparameter selection we performed 10-fold validation for splitting the training data into 90% training set and 10% validation test. We then choose the best performing set of hyperparameters on all the validation sets, retrain the models with the best set of hyperparameters on the entire training data and finally report the performance on the unseen test data. The performance of all kernels on each data set is shown in table 2.

For C-SVM we chose the cost function value

$$C \in \{0.01, 0.1, 1, 10, 100\}$$

and for each kernel we used the following hyperparameter sets

- RNTK: We only used single layer RNTK, we $\phi = \text{ReLU}$ and the following hyperparameter sets for the variances:

$$\sigma_w \in \{1.34, 1.35, 1.36, 1.37, 1.38, 1.39, 1.40, 1.41, 1.42, \sqrt{2}, 1.43, 1.44, 1.45, 1.46, 1.47\}$$
$$\sigma_u = 1$$
$$\sigma_b \in \{0, 0.01, 0.05, 0.1, 0.2, 0.3, 0.4, 0.5, 0.7, 0.9, 1, 2\}$$
$$\sigma_h \in \{0, 0.01, 0.1, 0.5, 1\}$$

- NTK: The formula for NTK of $L$-layer MLP (Jacot et al., 2018) for $\boldsymbol{x}, \boldsymbol{x}' \in \mathbb{R}^m$ is:

$$\Sigma^{(1)} = \frac{\sigma_w^2}{m}\langle \boldsymbol{x}, \boldsymbol{x}'\rangle + \sigma_b^2$$
$$\Sigma^{(\ell)}(\boldsymbol{x}, \boldsymbol{x}') = \sigma_w^2 \mathrm{V}_\phi[K^{(\ell)}(\boldsymbol{x}, \boldsymbol{x}')] + \sigma_b^2 \qquad\qquad \ell \in [L]$$
$$\dot{\Sigma}^{(\ell)}(\boldsymbol{x}, \boldsymbol{x}') = \sigma_w^2 \mathrm{V}_{\phi'}[K^{(\ell+1)}(\boldsymbol{x}, \boldsymbol{x}')] \qquad\qquad \ell \in [L]$$
$$\boldsymbol{K}^{(\ell)}(\boldsymbol{x}, \boldsymbol{x}') = \left[ \begin{array}{cc} \Sigma^{(\ell-1)}(\boldsymbol{x}, \boldsymbol{x}) & \Sigma^{(\ell-1)}(\boldsymbol{x}, \boldsymbol{x}') \\ \Sigma^{(\ell-1)}(\boldsymbol{x}, \boldsymbol{x}') & \Sigma^{(\ell-1)}(\boldsymbol{x}', \boldsymbol{x}') \end{array} \right]$$
$$\mathcal{K}(\boldsymbol{x}, \boldsymbol{x}') = \sigma_v^2 \mathrm{V}_\phi[K^{(L+1)}(\boldsymbol{x}, \boldsymbol{x}')]$$
$$k_{\text{NTK}} = \sum_{\ell=1}^{L}\left( \Sigma^{(\ell)}(\boldsymbol{x}, \boldsymbol{x}') \prod_{\ell'=\ell}^{L} \dot{\Sigma}^{(\ell)}(\boldsymbol{x}, \boldsymbol{x}') \right) + \mathcal{K}(\boldsymbol{x}, \boldsymbol{x}')$$

and we used the following hyperparamters

$$L \in [10]$$
$$\sigma_w \in \{0.5, 1, \sqrt{2}, 2, 2.5, 3\}$$
$$\sigma_b \in \{0, 0.01, 0.1, 0.2, 0.5, 0.8, 1, 2, 5\}$$

- RBF:

$$k_{\text{RBF}}(\boldsymbol{x}, \boldsymbol{x}') = e^{(-\alpha\|\boldsymbol{x}-\boldsymbol{x}'\|_2^2)}$$
$$\alpha \in \{0.01, 0.05, 0.1, 0.2, 0.5, 0.6, 0.7, 0.8, 1, 2, 3, 4, 5, 10, 20, 30, 40, 100\}$$

- Polynomial:

$$k_{\text{Polynomial}}(\boldsymbol{x}, \boldsymbol{x}') = (r + \langle \boldsymbol{x}, \boldsymbol{x}'\rangle)^d$$
$$d \in [5]$$
$$r \in \{0, 0.1, 0.2, 0.5, 1, 2\}$$

Table 2: Test accuracy of each model on 56 time series data set from UCR time-series classification data repository (Dau et al., 2019).

| Dataset | RNTK | NTK | RBF | POLY | Gaussian RNN | Identity RNN | GRU |
|---|---|---|---|---|---|---|---|
| Strawberry | **98.38** | 97.57 | 97.03 | 96.76 | 94.32 | 75.4 | 91.62 |
| ProximalPhalanxOutlineCorrect | **89** | 87.97 | 87.29 | 86.94 | 82.81 | 74.57 | 86.94 |
| PowerCons | 97.22 | 97.22 | 96.67 | 91.67 | 96.11 | 95 | **99.44** |
| Ham | 70.48 | **71.63** | 66.67 | 71.43 | 53.33 | 60 | 60.95 |
| SmallKitchenAppliances | 67.47 | 38.4 | 40.27 | 37.87 | 60.22 | **76** | 71.46 |
| ScreenType | 41.6 | 43.2 | **43.47** | 38.4 | 40 | 41.06 | 36.26 |
| MiddlePhalanxOutlineCorrect | 57.14 | 57.14 | 48.7 | 64.29 | **76.28** | 57.04 | 74.57 |
| RefrigerationDevices | 46.93 | 37.07 | 36.53 | 41.07 | 36 | **50.93** | 46.66 |
| Yoga | **84.93** | 84.63 | 84.63 | 84.87 | 46.43 | 76.66 | 61.83 |
| Computers | **59.2** | 55.2 | 58.8 | 56.4 | 53.2 | 55.2 | 58.8 |
| ECG5000 | 93.76 | **94.04** | 93.69 | 93.96 | 88.4 | 93.15 | 93.26 |
| Fish | **90.29** | 84 | 85.71 | 88 | 28 | 38.28 | 24 |
| UWaveGestureLibraryX | **79.59** | 78.7 | 78.48 | 65.83 | 55.97 | 75.34 | 73.64 |
| UWaveGestureLibraryY | **71.56** | 70.63 | 70.35 | 70.32 | 44.5 | 65.18 | 65.38 |
| UWaveGestureLibraryZ | **73.95** | 73.87 | 72.89 | 71.94 | 43.29 | 67.81 | 70.32 |
| StarLightCurves | 95.94 | **96.19** | 94.62 | 94.44 | 82.13 | 86.81 | 96.15 |
| CricketX | 60.51 | 59.49 | 62.05 | 62.56 | 8.46 | **63.58** | 26.41 |
| CricketY | **63.85** | 58.97 | 60.51 | 59.74 | 15.89 | 59.23 | 36.15 |
| CricketZ | **60.26** | 59.23 | 62.05 | 59.23 | 8.46 | 57.94 | 41.28 |
| DistalPhalanxOutlineCorrect | **77.54** | **77.54** | 75.36 | 73.91 | 69.92 | 69.56 | 75 |
| Worms | **57.14** | 50.65 | 55.84 | 50.65 | 35.06 | 49.35 | 41.55 |
| SyntheticControl | 98.67 | 96.67 | 98 | 97.67 | 92.66 | 97.66 | **99** |
| Herring | 56.65 | **59.38** | **59.38** | **59.38** | 23.28 | 59.37 | **59.38** |
| MedicalImages | 74.47 | 73.29 | **75.26** | 74.61 | 48.15 | 64.86 | 69.07 |
| SwedishLeaf | 90.56 | 91.04 | **91.36** | 90.72 | 59.2 | 45.92 | 91.04 |
| ChlorineConcentration | 90.76 | 77.27 | 86.35 | **91.54** | 65.99 | 55.75 | 61.14 |
| SmoothSubspace | **96** | 87.33 | 92 | 86.67 | 94 | 95.33 | 92.66 |
| TwoPatterns | 94.25 | 90.45 | 91.25 | 93.88 | 99.7 | 99.9 | **100** |
| Faceall | 74.14 | **83.33** | 83.25 | 82.43 | 53.66 | 70.53 | 70.65 |
| DistalPhalanxTW | 66.19 | **69.78** | 66.91 | 67.37 | 67.62 | 64.74 | 69.06 |
| MiddlePhalanxTW | 57.79 | **61.04** | 59.74 | 60.39 | 58.44 | 58.44 | 59.09 |
| FacesUCR | 81.66 | 80.2 | 80.34 | **82.98** | 53.21 | 75.26 | 79.46 |
| OliveOil | **90** | 86.67 | 86.67 | 83.33 | 66.66 | 40 | 40 |
| UMD | 91.67 | 92.36 | 97.22 | 90.97 | 44.44 | 71.52 | **100** |
| nsectEPGRegular | 99.6 | 99.2 | 99.6 | 96.79 | **100** | **100** | 98.39 |
| Meat | **93.33** | **93.33** | **93.33** | **93.33** | 0.55 | 55 | 33.33 |
| Lightning2 | **78.69** | 73.77 | 70.49 | 68.85 | 45.9 | 70.49 | 67.21 |
| Lightning7 | 61.64 | 60.27 | 63.01 | 60.27 | 23.28 | 69.86 | **76.71** |
| Car | **83.33** | 78.83 | 80 | 80 | 23.33 | 58.33 | 26.66 |
| GunPoint | **98** | 95.33 | 95.33 | 94 | 82 | 74.66 | 80.66 |
| Arrowhead | 80.57 | **83.43** | 80.57 | 74.86 | 48 | 56 | 37.71 |
| Coffee | **100** | **100** | 92.86 | 92.86 | **100** | 42.85 | 57.14 |
| Trace | 96 | 81 | 76 | 76 | 70 | 71 | **100** |
| ECG200 | **93** | 89 | 89 | 86 | 86 | 72 | 76 |
| plane | **98.1** | 96.19 | 97.14 | 97.14 | 96.19 | 84.76 | 96.19 |
| GunPointOldVersusYoung | **98.73** | 97.46 | **98.73** | 94.6 | 53.96 | 52.38 | 98.41 |
| GunPointMaleVersusFemale | 99.05 | **99.68** | 99.37 | **99.68** | 68.67 | 52.53 | 99.68 |
| GunPointAgeSpan | **96.52** | 94.62 | 95.89 | 93.99 | 47.78 | 47.78 | 95.56 |
| FreezerRegularTrain | **97.44** | 94.35 | 96.46 | 96.84 | 76.07 | 7.5 | 86.59 |
| SemgHandSubjectCh2 | 84.22 | 85.33 | 86.14 | 86.67 | 20 | 36.66 | **89.11** |
| WormsTwoClass | **62.34** | **62.34** | 61.04 | 59.74 | 51.94 | 46.75 | 57.14 |
| Earthquakes | 74.82 | 74.82 | 74.82 | 74.82 | 65.46 | **76.97** | **76.97** |
| FiftyWords | 68.57 | 68.57 | **69.67** | 68.79 | 34.28 | 60.21 | 65.27 |
| Beef | 90 | 73.33 | 83.33 | **93.33** | 26.67 | 46.67 | 36.67 |
| Adiac | 76.63 | 71.87 | 73.40 | **77.75** | 51.4 | 16.88 | 60.61 |
| WordSynonyms | 57.99 | 58.46 | 61.13 | **62.07** | 17.71 | 45.77 | 53.76 |

**Finite-width RNN settings.** We used 3 different RNNs. The first is a ReLU RNN with Gaussian initialization with the same NTK initialization scheme, where parameter variances are $\sigma_w = \sigma_v = \sqrt{2}$,

$\sigma_u = 1$ and $\sigma_b = 0$. The second is a ReLU RNN with identity initialization following (Le et al., 2015). The third is a GRU (Cho et al., 2014) with uniform initialization. All models are trained with RMSProp algorithm for 200 epochs. Early stopping is implemented when the validation set accuracy does not improve for 5 consecutive epochs.

We perform standard 5-fold cross validation. For each RNN architecture we used hyperparamters of number of layer, number of hidden units and learning rate as

$$L \in \{1, 2\}$$
$$n \in \{50, 100, 200, 500\}$$
$$\eta \in \{0.01, 0.001, 0.0001, 0.00001\}$$

**Metrics descriptions** First, only in this paragraph, let $i \in \{1, 2, ..., N\}$ index a total of $N$ datasets and $j \in \{1, 2, ..., M\}$ index a total of $M$ classifiers. Let $y_{ij}$ be the accuracy of the $j$-th classifer on the $i$-th dataset. We reported results on 4 metrics: average accuracy (Acc. mean), P90, P95, PMA and Friedman Rank. P90 and P95 is the fraction of datasets that the classifier achieves at least 90% and 95% of the maximum achievable accuracy for each dataset, i.e.,

$$P90_j = \frac{1}{N} \sum_i \mathbf{1}(y_{ij} \geq 0.9(\max_j y_{ij})). \tag{21}$$

PMA is the accuracy of the classifier on a dataset divided by the maximum achievable accuracy on that dataset, averaged over all datasets:

$$\text{PMA}_j = \frac{1}{N} \sum_i \frac{y_{ij}}{\max_j y_{ij}}. \tag{22}$$

Friedman Rank (Fernández-Delgado et al., 2019) first ranks the accuracy of each classifier on each dataset and then takes the average of the ranks for each classifier over all datasets, i.e.,

$$\text{FR}_j = \frac{1}{N} \sum_i r_{ij}, \tag{23}$$

where $r_{ij}$ is the ranking of the $j$-th classifier on the $i$-th dataset.

Note that a better classifier achieves a lower Friedman Rank, Higher P/90 and PMA.

**Remark.** In order to provide insight into the performance of RNTK in long time steps setting, we picked two datasets with more that 1000 times steps: SemgHandSubjectCh2 ($T = 1024$) and StarLightCurves ($T = 1024$).

## A.2 TIME SERIES REGRESSION

For time series regression, we used the 5-fold validation of training set and same hyperparamter sets for all kernels. For training we kernel ridge regression with ridge term chosen form

$$\lambda \in \{0, 0.01, 0.1, 0.2, 0.3, 0.4, 0.5, 0.6, 0.7, 0.8, 0.8, 1, 2, 3, 4, 5, 6, 7, 8, 10, 100\}$$

## B PROOFS FOR THEOREMS 1 AND 3: RNTK CONVERGENCE AT INITIALIZATION

### B.1 PRELIMINARY: NETSOR PROGRAMS

Calculation of NTK in any architecture relies on finding the GP kernels that correspond to each pre-activation and gradient layers at initialization. For feedforward neural networks with $n_1, \ldots, n_L$ number of neurons (channels in CNNs) at each layer the form of this GP kernels can be calculated via taking the limit of $n_1, \ldots, n_L$ sequentially one by one. The proof is given by induction, where by conditioning on the previous layers, each entry of the current layer is sum of infinite i.i.d Gaussian random variables, and based on Central Limit Theorem (CLT), it becomes a Gaussian process with

kernel calculated based on the previous layers. Since the first layer is an affine transformation of input with Gaussian weights, it is a Gaussian process and the proof is completed. See (Lee et al., 2018; Duvenaud et al., 2014; Novak et al., 2019; Garriga-Alonso et al., 2019) for a formal treatment. However, due to weight-sharing, sequential limit is not possible and condoning on previous layers does not result in i.i.d. weights. Hence the aforementioned arguments break. To deal with it, in (Yang, 2019a) a proof using Gaussian conditioning trick (Bolthausen, 2014) is presented which allows use of recurrent weights in a network. More precisely, it has been demonstrated than neural networks (without batch normalization) can be expressed and a series of matrix multiplication and (piece wise) nonlinearity application, generally referred as *Netsor programs*. It has been shown that any architecture that can be expressed as *Netsor programs* that converge to GPs as width goes to infinity in the same rate, which a general rule to obtain the GP kernels. For completeness of this paper, we briefly restate the results from (Yang, 2019a) which we will use later for calculation derivation of RNTK.

There are 3 types of variables in *Netsor programs*; $A$-vars, $G$-vars and $H$-vars. $A$-vars are matrices and vectors with i.i.d Gaussian entries, $G$-vars are vectors introduced by multiplication of a vector by an $A$-var and $H$-vars are vectors after coordinate wise nonlinearities is applied to $G$-vars. Generally, $G$-vars can be thought of as pre-activation layers which are asymptotically treated as a Gaussian distributed vectors, $H$-vars as after-activation layers and $A$-vars are the weights. Since in neural networks inputs are immediately multiplied by a weight matrix, it can be thought of as an $G$-var, namely $\boldsymbol{g}_{in}$. Generally *Netsor programs* supports $G$-vars with different dimension, however the asymptotic behavior of a neural networks described by *Netsor programs* does not change under this degree of freedom, as long as they go to infinity at the same rate. For simplicity, let the $G$-vars and $H$-vars have the same dimension $n$ since the network of interest is RNN and all pre-activation layers have the same dimension. We introduce the *Netsor programs* under this simplification. To produce the output of a neural network, *Netsor programs* receive a set of $G$-vars and $A$-vars as input, and new variables are produced sequentially using the three following operators:

- `Matmul` : multiplication of an $A$-var: $\boldsymbol{A}$ with an $H$-var: $\boldsymbol{h}$, which produce a new $G$-var, $\boldsymbol{g}$.

$$\boldsymbol{g} = \boldsymbol{A}\boldsymbol{h} \tag{24}$$

- `Lincomp`: Linear combination of $G$-vars, $\boldsymbol{g}^i$, $1 \le i \le k$ , with coefficients $a^i \in \mathbb{R}$ $1 \le i \le k$ which produce of new $G$-var:

$$\boldsymbol{g} = \sum_{i=1}^{k} a^i \boldsymbol{g}^i \tag{25}$$

- `Nonlin`: creating a new $H$-var, $\mathbf{h}$, by using a nonlinear function $\phi : \mathbb{R}^k \to \mathbb{R}$ that act coordinate wise on a set of $G$-vars, $\boldsymbol{g}^i$, $1 \le i \le k$ :

$$\boldsymbol{h} = \varphi(\boldsymbol{g}^1, \dots, \boldsymbol{g}^k) \tag{26}$$

Any output of the neural network $y \in \mathbb{R}$ should be expressed as inner product of a new $A$-var which has not been used anywhere else in previous computations and an $H$-var:

$$y = \boldsymbol{v}^\top \boldsymbol{h}$$

Any other output can be produced by another $\boldsymbol{v}'$ and $\boldsymbol{h}'$ (possibility the same $\boldsymbol{h}$ or $\boldsymbol{v}$).

It is assumed that each entry of any $A$-var : $\boldsymbol{A} \in \mathbb{R}^{n \times n}$ in the *netsor programs* computations is drawn from $\mathcal{N}(0, \frac{\sigma_a^2}{n})$ and the input $G$-vars are Gaussian distributed. The collection of a specific entry of all $G$-vars of in the *netsor program* converges in probability to a Gaussian vector $\{[\boldsymbol{g}^1]_i, \dots, [\boldsymbol{g}^k]_i\} \sim \mathcal{N}(\boldsymbol{\mu}, \boldsymbol{\Sigma})$ for all $i \in [n]$ as $n$ goes to infinity.

Let $\mu(\boldsymbol{g}) := \mathbb{E}\big[[\boldsymbol{g}]_i\big]$ be the mean of a $G$-var and $\Sigma(\boldsymbol{g}, \boldsymbol{g}') := \mathbb{E}\big[[\boldsymbol{g}]_i \cdot [\boldsymbol{g}']_i\big]$ be the covariance between any two $G$-vars. The general rule for $\mu(\boldsymbol{g})$ is given by the following equations:

$$\mu(\boldsymbol{g}) = \begin{cases} \mu^{in}(\boldsymbol{g}) & \text{if } \boldsymbol{g} \text{ is input} \\ \sum_{i=1}^{k} a^i \mu(\boldsymbol{g}^i) & \text{if } \boldsymbol{g} = \sum_{i=1}^{k} a^i \boldsymbol{g}^i \\ 0 & \text{otherwise} \end{cases} \tag{27}$$

For $g$ and $g'$, let $\mathcal{G} = \{g^1, \ldots, g^r\}$ be the set of $G$-vars that has been introduced *before* $g$ and $g'$ with distribution $\mathcal{N}(\boldsymbol{\mu}_{\mathcal{G}}, \boldsymbol{\Sigma}_{\mathcal{G}})$, where $\boldsymbol{\Sigma}_{\mathcal{G}} \in \mathbb{R}^{|\mathcal{G}| \times |\mathcal{G}|}$ containing the pairwise covariances between the $G$-vars. $\Sigma(g, g')$ is calculated via the following rules:

$$\Sigma(\boldsymbol{g}, \boldsymbol{g}') = \begin{cases} \Sigma^{\text{in}}(\boldsymbol{g}, \boldsymbol{g}') & \text{if } \boldsymbol{g} \text{ and } \boldsymbol{g}' \text{ are inputs} \\ \sum_{i=1}^{k} a^i \Sigma(\boldsymbol{g}^i, \boldsymbol{g}') & \text{if } \boldsymbol{g} = \sum_{i=1}^{k} a^i \boldsymbol{g}^i \\ \sum_{i=1}^{k} a^i \Sigma(\boldsymbol{g}, \boldsymbol{g}^i) & \text{if } \boldsymbol{g}' = \sum_{i=1}^{k} a^i \boldsymbol{g}^i \\ \sigma_A^2 \underset{\mathbf{z} \sim \mathcal{N}(\boldsymbol{\mu}, \boldsymbol{\Sigma}_{\mathcal{G}})}{\mathbb{E}} [\varphi(\mathbf{z}) \bar{\varphi}(\mathbf{z})] & \text{if } \boldsymbol{g} = \boldsymbol{A}\boldsymbol{h} \text{ and } \boldsymbol{g}' = \boldsymbol{A}\boldsymbol{h}' \\ 0 & \text{otherwise} \end{cases} \tag{28}$$

Where $\boldsymbol{h} = \varphi(g^1, \ldots, g^r)$ and $\boldsymbol{h}' = \bar{\varphi}(g^1, \ldots, g^r)$ are functions of $G$-vars in $\mathcal{G}$ from possibly different nonlinearities. This set of rules presents a recursive method for calculating the GP kernels in a network where the recursive formula starts from data dependent quantities $\Sigma^{\text{in}}$ and $\mu^{in}$ which are given.

All the above results holds when the nonlinearities are bounded uniformly by $e^{(cx^{2-\alpha})}$ for some $\alpha > 0$ and when their derivatives exist.

**Standard vs. NTK initialization.** The common practice (which *netsor programs* uses) is to initialize DNNs weights $[\boldsymbol{A}]_{i,j}$ with $\mathcal{N}(0, \frac{\sigma_a}{\sqrt{n}})$ (known as *standard* initialization) where generally $n$ is the number of units in the previous layer. In this paper we have used a different parameterization scheme as used in (Jacot et al., 2018) and we factor the standard deviation as shown in 3 and initialize weights with standard standard Gaussian. This approach does not change the the forward computation of DNN, but normalizes the backward computation (when computing the gradients) by factor $\frac{1}{n}$, otherwise RNTK will be scales by $n$. However this problem can be solved by scaling the step size by $\frac{1}{n}$ and there is no difference between *NTK* and *standard* initialization (Lee et al., 2019).

### B.2 PROOF FOR THEOREM 1: SINGLE LAYER CASE

We first derive the RNTK in a simpler setting, i.e., a single layer and single output RNN. We then generalize the results to multi-layer and multi-output RNNs. We drop the layer index $\ell$ to simplify notation. From 3 and 4, the forward pass for computing the output under NTK initialization for each input $\boldsymbol{x} = \{\boldsymbol{x}_t\}_{t=1}^{T}$ is given by:

$$\boldsymbol{g}^{(t)}(\boldsymbol{x}) = \frac{\sigma_w}{\sqrt{m}} \mathbf{W} \boldsymbol{h}^{(t-1)}(\boldsymbol{x}) + \frac{\sigma_u}{\sqrt{n}} \mathbf{U} \boldsymbol{x}_t + \sigma_b \mathbf{b} \tag{29}$$

$$\boldsymbol{h}^{(t)}(\boldsymbol{x}) = \phi\left(\boldsymbol{g}^{(t)}(\boldsymbol{x})\right) \tag{30}$$

$$f_\theta(\boldsymbol{x}) = \frac{\sigma_v}{\sqrt{n}} \boldsymbol{v}^\top \boldsymbol{h}^{(T)}(\boldsymbol{x}) \tag{31}$$

Note that (29), (30) and (31) use all the introduced operators introduced in 24, 25 and 26 given input variables $\mathbf{W}, \{\mathbf{U}\boldsymbol{x}_t\}_{t=1}^{T}, \mathbf{b}, \boldsymbol{v}$ and $\boldsymbol{h}^{(0)}(\boldsymbol{x})$.

First, we compute the kernels of forward pass $\Sigma^{(t,t')}(\boldsymbol{x}, \boldsymbol{x}')$ and backward pass $\Pi^{(t,t')}(\boldsymbol{x}, \boldsymbol{x}')$ introduced in (6) and (7) for two input $\boldsymbol{x}$ and $\boldsymbol{x}'$. Note that based on (27) the mean of all variables is zero since the inputs are all zero mean. In the forward pass for the intermediate layers we have:

$$\Sigma^{(t,t')}(\boldsymbol{x}, \boldsymbol{x}') = \Sigma(\boldsymbol{g}^{(t)}(\boldsymbol{x}), \boldsymbol{g}^{(t')}(\boldsymbol{x}'))$$

$$= \Sigma\left(\frac{\sigma_w}{\sqrt{n}} \mathbf{W} \boldsymbol{h}^{(t-1)}(\boldsymbol{x}) + \frac{\sigma_u}{\sqrt{m}} \mathbf{U} \boldsymbol{x}_t + \sigma_b \mathbf{b}, \frac{\sigma_w}{\sqrt{n}} \mathbf{W} \boldsymbol{h}^{(t'-1)}(\boldsymbol{x}') + \frac{\sigma_u}{\sqrt{m}} \mathbf{U} \boldsymbol{x}'_{t'} + \sigma_b \mathbf{b}\right)$$

$$= \Sigma\left(\frac{\sigma_w}{\sqrt{n}} \mathbf{W} \boldsymbol{h}^{(t-1)}(\boldsymbol{x}), \frac{\sigma_w}{\sqrt{n}} \mathbf{W} \boldsymbol{h}^{(t'-1)}(\boldsymbol{x}')\right) + \Sigma^{\text{in}}\left(\frac{\sigma_u}{\sqrt{m}} \mathbf{U} \boldsymbol{x}_t, \frac{\sigma_u}{\sqrt{m}} \mathbf{U} \boldsymbol{x}'_{t'}\right) + \Sigma^{\text{in}}\left(\sigma_b \mathbf{b}, \sigma_b \mathbf{b}\right).$$

We have used the second and third rule in (28) to expand the formula, We have also used the first and fifth rule to set the cross term to zero, i.e.,

$$\Sigma\left(\frac{\sigma_w}{\sqrt{n}}\mathbf{W}\boldsymbol{h}^{(t-1)}(\boldsymbol{x}), \frac{\sigma_u}{\sqrt{n}}\mathbf{U}\boldsymbol{x}'_{t'}\right) = 0$$

$$\Sigma\left(\frac{\sigma_w}{\sqrt{n}}\mathbf{W}\boldsymbol{h}^{(t-1)}(\boldsymbol{x}), \sigma_b\mathbf{b}\right) = 0$$

$$\Sigma\left(\frac{\sigma_u}{\sqrt{m}}\mathbf{U}\boldsymbol{x}_t, \frac{\sigma_w}{\sqrt{n}}\mathbf{W}\boldsymbol{h}^{(t'-1)}(\boldsymbol{x}')\right) = 0$$

$$\Sigma\left(\sigma_b\mathbf{b}, \frac{\sigma_w}{\sqrt{n}}\mathbf{W}\boldsymbol{h}^{(t'-1)}(\boldsymbol{x}')\right) = 0$$

$$\Sigma^{\text{in}}\left(\frac{\sigma_u}{\sqrt{m}}\mathbf{U}\boldsymbol{x}_t, \sigma_b\mathbf{b}\right) = 0$$

$$\Sigma^{\text{in}}\left(\sigma_b\mathbf{b}, \frac{\sigma_u}{\sqrt{m}}\mathbf{U}\boldsymbol{x}'_{t'}\right) = 0.$$

For the non-zero terms we have

$$\Sigma^{\text{in}}\left(\sigma_b\boldsymbol{b}, \sigma_b\boldsymbol{b}\right) = \sigma_b^2$$

$$\Sigma^{\text{in}}\left(\frac{\sigma_u}{\sqrt{m}}\mathbf{U}\boldsymbol{x}_t, \frac{\sigma_u}{\sqrt{m}}\mathbf{U}\boldsymbol{x}'_{t'}\right) = \frac{\sigma_u^2}{m}\langle\boldsymbol{x}_t, \boldsymbol{x}'_{t'}\rangle,$$

which can be achieved by straight forward computation. If $t \neq 1$ and $t' \neq 1$, by using the forth rule in (28) we have

$$\Sigma\left(\frac{\sigma_w}{\sqrt{n}}\mathbf{W}\boldsymbol{h}^{(t-1)}(\boldsymbol{x}), \frac{\sigma_w}{\sqrt{n}}\mathbf{W}\boldsymbol{h}^{(t'-1)}(\boldsymbol{x}')\right) = \sigma_w^2 \underset{\mathbf{z}\sim\mathcal{N}(0,\boldsymbol{K}^{(t,t')}(\boldsymbol{x},\boldsymbol{x}'))}{\mathbb{E}} [\phi(\mathbf{z}_1)\phi(\mathbf{z}_2)] = \mathrm{V}_\phi\left[\boldsymbol{K}^{(t,t')}(\boldsymbol{x}, \boldsymbol{x}')\right].$$

With $\boldsymbol{K}^{(t,t')}(\boldsymbol{x}, \boldsymbol{x}')$ defined in (16). Otherwise, it will be zero by the fifth rule (if $t$ or $t = 1$) .

Here the set of previously introduced $G$-vars is $\mathcal{G} = \left\{\{\boldsymbol{g}^{(\alpha)}(\boldsymbol{x})\}, \mathbf{U}\boldsymbol{x}_\alpha\}_{\alpha=1}^{t-1}, \{\boldsymbol{g}^{(\alpha')}(\boldsymbol{x}'), \mathbf{U}\boldsymbol{x}'_{\alpha'}\}_{\alpha'=1}^{t'-1}, \boldsymbol{h}^{(0)}(\boldsymbol{x}), \boldsymbol{h}^{(0)}(\boldsymbol{x}')\right\}$, but the dependency is only on the last layer $G$-vars, $\varphi(\{\boldsymbol{g} : \boldsymbol{g} \in \mathcal{G}\}) = \phi(\boldsymbol{g}^{(t-1)}(\boldsymbol{x}))$, $\bar{\varphi}((\{\boldsymbol{g} : \boldsymbol{g} \in \mathcal{G}\})) = \phi(\boldsymbol{g}^{(t'-1)}(\boldsymbol{x}'))$, leading the calculation to the operator defined in (10). As a result

$$\Sigma^{(t,t')}(\boldsymbol{x}, \boldsymbol{x}') = \sigma_w^2 \mathrm{V}_\phi\left[\boldsymbol{K}^{(t,t')}(\boldsymbol{x}, \boldsymbol{x}')\right] + \frac{\sigma_u^2}{m}\langle\boldsymbol{x}_t, \boldsymbol{x}'_{t'}\rangle + \sigma_b^2.$$

To complete the recursive formula, using the same procedure for the first layers we have

$$\Sigma^{(1,1)}(\boldsymbol{x}, \boldsymbol{x}') = \sigma_w^2\sigma_h^2 1_{(\boldsymbol{x}=\boldsymbol{x}')} + \frac{\sigma_u^2}{m}\langle\boldsymbol{x}_1, \boldsymbol{x}'_1\rangle + \sigma_b^2,$$

$$\Sigma^{(1,t')}(\boldsymbol{x}, \boldsymbol{x}') = \frac{\sigma_u^2}{m}\langle\boldsymbol{x}_1, \boldsymbol{x}'_{t'}\rangle + \sigma_b^2,$$

$$\Sigma^{(t,1)}(\boldsymbol{x}, \boldsymbol{x}') = \frac{\sigma_u^2}{m}\langle\boldsymbol{x}_t, \boldsymbol{x}'_1\rangle + \sigma_b^2.$$

The output GP kernel is calculated via

$$\mathcal{K}(\boldsymbol{x}, \boldsymbol{x}') = \sigma_v^2 \mathrm{V}_\phi\left[\boldsymbol{K}^{(T+1,T'+1)}(\boldsymbol{x}, \boldsymbol{x}')\right]$$

The calculation of the gradient vectors $\boldsymbol{\delta}^{(t)}(\boldsymbol{x}) = \sqrt{n}\left(\nabla_{\boldsymbol{g}^{(t)}(\boldsymbol{x})} f_\theta(\boldsymbol{x})\right)$ in the backward pass is given by

$$\boldsymbol{\delta}^{(T)}(\boldsymbol{x}) = \sigma_v \boldsymbol{v} \odot \phi'(\boldsymbol{g}^{(T)}(\boldsymbol{x}))$$

$$\boldsymbol{\delta}^{(t)}(\boldsymbol{x}) = \frac{\sigma_w}{\sqrt{n}}\mathbf{W}^\top\left(\phi'(\boldsymbol{g}^{(t)}(\boldsymbol{x})) \odot \boldsymbol{\delta}^{(t+1)}(\boldsymbol{x})\right) \qquad t \in [T-1]$$

To calculate the backward pass kernels, we rely on the following Corollary from (Yang, 2020b)

**Corollary 1** *In infinitely wide neural networks weights used in calculation of back propagation gradients ($\mathbf{W}^\top$) is an i.i.d copy of weights used in forward propagation ($\mathbf{W}$) as long as the last layer weight ($\boldsymbol{v}$) is sampled independently from other parameters and has mean 0.*

The immediate result of Corollary 1 is that $\boldsymbol{g}^{(t)}(\boldsymbol{x})$ and $\boldsymbol{\delta}^{(t)}(\boldsymbol{x})$ are two independent Gaussian vector as their covariance is zero based on the fifth rule in (28). Using this result, we have:

$$
\begin{aligned}
\Pi^{(t,t')}(\boldsymbol{x},\boldsymbol{x}') &= \Sigma\left(\boldsymbol{\delta}^{(t)}(\boldsymbol{x}),\boldsymbol{\delta}^{(t')}(\boldsymbol{x})\right) \\
&= \mathbb{E}\left[[\boldsymbol{\delta}^{(t)}(\boldsymbol{x})]_i \cdot [\boldsymbol{\delta}^{(t')}(\boldsymbol{x}')]_i\right] \\
&= \sigma_w^2 \mathbb{E}\left[[\phi'(\boldsymbol{g}^{(t)}(\boldsymbol{x}))]_i \cdot [\boldsymbol{\delta}^{(t+1)}(\boldsymbol{x})]_i \cdot [\phi'(\boldsymbol{g}^{(t')}(\boldsymbol{x}'))]_i \cdot [\boldsymbol{\delta}^{(t'+1)}(\boldsymbol{x}')]_i\right] \\
&= \sigma_w^2 \underset{\mathbf{z}\sim\mathcal{N}(0,\boldsymbol{K}^{(t+1,t+1')}(\boldsymbol{x},\boldsymbol{x}'))}{\mathbb{E}}[\phi'(\mathbf{z}_1)\cdot\phi'(\mathbf{z}_2)]\cdot\mathbb{E}\left[[\boldsymbol{\delta}^{(t+1)}(\boldsymbol{x})]_i\cdot[\boldsymbol{\delta}^{(t'+1)}(\boldsymbol{x}')]_i\right] \\
&= \sigma_w^2 \mathrm{V}_{\phi'}\left[\boldsymbol{K}^{(t+1,t'+1)}(\boldsymbol{x},\boldsymbol{x}')\right]\Pi^{(t+1,t'+1)}(\boldsymbol{x},\boldsymbol{x}').
\end{aligned}
$$

If $T'-t'=T-t$, then the the formula will lead to

$$
\begin{aligned}
\Pi^{(T,T')}(\boldsymbol{x},\boldsymbol{x}') &= \mathbb{E}\left[[\boldsymbol{\delta}^{(T)}(\boldsymbol{x})]_i,[\boldsymbol{\delta}^{(T')}(\boldsymbol{x}')]_i\right] \\
&= \sigma_v^2 \mathbb{E}\left[[\boldsymbol{v}]_i\cdot[\phi'(\boldsymbol{g}^{(T)}(\boldsymbol{x}))]_i\cdot[\boldsymbol{v}]_i\cdot[\phi'(\boldsymbol{g}^{(T')}(\boldsymbol{x}'))]_i\right] \\
&= \mathbb{E}\left[[\phi'(\boldsymbol{g}^{(T)}(\boldsymbol{x}))]_i\cdot[\phi'(\boldsymbol{g}^{(T')}(\boldsymbol{x}'))]_i\right]\cdot\mathbb{E}\left[[\boldsymbol{v}]_i[\boldsymbol{v}]_i\right] \\
&= \sigma_v^2\mathrm{V}_{\phi'}\left[\boldsymbol{K}^{(T+1,T+\tau+1)}(\boldsymbol{x},\boldsymbol{x}')\right].
\end{aligned}
$$

Otherwise it will end to either of two cases for some $t''<T$ or $T'$ and by the fifth rule in (28) we have:

$$
\Sigma\left(\boldsymbol{\delta}^{(t'')}(\boldsymbol{x}),\boldsymbol{\delta}^{(T')}(\boldsymbol{x})\right) = \Sigma\left(\frac{\sigma_w}{\sqrt{n}}\mathbf{W}^\top\left(\phi'(\boldsymbol{g}^{(t'')}(\boldsymbol{x}))\odot\boldsymbol{\delta}^{(t''+1)}(\boldsymbol{x}')\right),\boldsymbol{v}\odot\phi'(\boldsymbol{g}^{(T')}(\boldsymbol{x}))\right) = 0
$$

$$
\Sigma\left(\boldsymbol{\delta}^{(T)}(\boldsymbol{x}),\boldsymbol{\delta}^{(t'')}(\boldsymbol{x})\right) = \Sigma\left(\boldsymbol{v}\odot\phi'(\boldsymbol{g}^{(T)}(\boldsymbol{x})),\frac{\sigma_w}{\sqrt{n}}\mathbf{W}^\top\left(\phi'(\boldsymbol{g}^{(t'')}(\boldsymbol{x}'))\odot\boldsymbol{\delta}^{(t''+1)}(\boldsymbol{x}')\right)\right) = 0.
$$

Without loss of generality, from now on assume $T'<T$ and $T'-T=\tau$, the final formula for computing the backward gradients becomes:

$$
\begin{aligned}
\Pi^{(T,T+\tau)}(\boldsymbol{x},\boldsymbol{x}') &= \sigma_v^2\mathrm{V}_{\phi'}\left[\boldsymbol{K}^{(T+1,T+\tau+1)}(\boldsymbol{x},\boldsymbol{x}')\right] \\
\Pi^{(t,t+\tau)}(\boldsymbol{x},\boldsymbol{x}') &= \sigma_w^2\mathrm{V}_{\phi'}\left[\boldsymbol{K}^{(t+1,t+\tau+1)}(\boldsymbol{x},\boldsymbol{x}')\right]\Pi^{(t+1,t+1+\tau)}(\boldsymbol{x},\boldsymbol{x}') \qquad t\in[T-1] \\
\Pi^{(t,t')}(\boldsymbol{x},\boldsymbol{x}') &= 0 \qquad\qquad\qquad\qquad\qquad\qquad\qquad\qquad\qquad t'-t\neq\tau
\end{aligned}
$$

Now we have derived the single layer RNTK. Recall that $\theta=\mathrm{Vect}\left[\{\mathbf{W},\mathbf{U},\mathbf{b},\boldsymbol{v}\}\right]$ contains all of the network's learnable parameters. As a result, we have:

$$
\nabla_\theta f_\theta(\boldsymbol{x}) = \mathrm{Vect}\left[\{\frac{\partial f_\theta(\boldsymbol{x})}{\partial\mathbf{W}},\frac{\partial f_\theta(\boldsymbol{x})}{\partial\mathbf{U}},\frac{\partial f_\theta(\boldsymbol{x})}{\partial\mathbf{b}},\frac{\partial f_\theta(\boldsymbol{x})}{\partial\boldsymbol{v}}\}\right].
$$

As a result

$$
\begin{aligned}
\langle\nabla_\theta f_\theta(\boldsymbol{x}),\nabla_\theta f_\theta(\boldsymbol{x}')\rangle &= \left\langle\frac{\partial f_\theta(\boldsymbol{x})}{\partial\mathbf{W}},\frac{\partial f_\theta(\boldsymbol{x}')}{\partial\mathbf{W}}\right\rangle + \left\langle\frac{\partial f_\theta(\boldsymbol{x})}{\partial\mathbf{U}},\frac{\partial f_\theta(\boldsymbol{x}')}{\partial\mathbf{U}}\right\rangle + \left\langle\frac{\partial f_\theta(\boldsymbol{x})}{\partial\mathbf{b}},\frac{\partial f_\theta(\boldsymbol{x}')}{\partial\mathbf{b}}\right\rangle \\
&\quad + \left\langle\frac{\partial f_\theta(\boldsymbol{x})}{\partial\boldsymbol{v}},\frac{\partial f_\theta(\boldsymbol{x}')}{\partial\boldsymbol{v}}\right\rangle
\end{aligned}
$$

Where the gradients of output with respect to weights can be formulated as the following compact form:

$$\frac{\partial f_\theta(\boldsymbol{x})}{\partial \mathbf{W}} = \sum_{t=1}^{T} \left(\frac{1}{\sqrt{n}}\boldsymbol{\delta}^{(t)}(\boldsymbol{x})\right) \cdot \left(\frac{\sigma_w}{\sqrt{n}}\boldsymbol{h}^{(t-1)}(\boldsymbol{x})\right)^\top$$

$$\frac{\partial f_\theta(\boldsymbol{x})}{\partial \mathbf{U}} = \sum_{t=1}^{T} \left(\frac{1}{\sqrt{n}}\boldsymbol{\delta}^{(t)}(\boldsymbol{x})\right) \cdot \left(\frac{\sigma_u}{\sqrt{m}}\boldsymbol{x}_t\right)^\top$$

$$\frac{\partial f_\theta(\boldsymbol{x})}{\partial \mathbf{b}} = \sum_{t=1}^{T} \left(\frac{\sigma_b}{\sqrt{n}}\boldsymbol{\delta}^{(t)}(\boldsymbol{x})\right)$$

$$\frac{\partial f_\theta(\boldsymbol{x})}{\partial \boldsymbol{v}} = \frac{\sigma_v}{\sqrt{n}}\boldsymbol{h}^{(T)}(\boldsymbol{x}).$$

As a result we have:

$$\left\langle \frac{\partial f_\theta(\boldsymbol{x})}{\partial \mathbf{W}}, \frac{\partial f_\theta(\boldsymbol{x}')}{\partial \mathbf{W}} \right\rangle = \sum_{t'=1}^{T'}\sum_{t=1}^{T} \left(\frac{1}{n}\left\langle \boldsymbol{\delta}^{(t)}(\boldsymbol{x}), \boldsymbol{\delta}^{(t')}(\boldsymbol{x}')\right\rangle\right) \cdot \left(\frac{\sigma_w^2}{n}\left\langle \boldsymbol{h}^{(t-1)}(\boldsymbol{x}), \boldsymbol{h}^{(t'-1)}(\boldsymbol{x}')\right\rangle\right)$$

$$\left\langle \frac{\partial f_\theta(\boldsymbol{x})}{\partial \mathbf{U}}, \frac{\partial f_\theta(\boldsymbol{x}')}{\partial \mathbf{U}} \right\rangle = \sum_{t'=1}^{T'}\sum_{t=1}^{T} \left(\frac{1}{n}\left\langle \boldsymbol{\delta}^{(t)}(\boldsymbol{x}), \boldsymbol{\delta}^{(t')}(\boldsymbol{x}')\right\rangle\right) \cdot \left(\frac{\sigma_u^2}{m}\left\langle \boldsymbol{x}_t, \boldsymbol{x}'_{t'}\right\rangle\right)$$

$$\left\langle \frac{\partial f_\theta(\boldsymbol{x})}{\partial \mathbf{b}}, \frac{\partial f_\theta(\boldsymbol{x}')}{\partial \mathbf{b}} \right\rangle = \sum_{t'=1}^{T'}\sum_{t=1}^{T} \left(\frac{1}{n}\left\langle \boldsymbol{\delta}^{(t)}(\boldsymbol{x}), \boldsymbol{\delta}^{(t')}(\boldsymbol{x}')\right\rangle\right) \cdot \sigma_b^2$$

$$\left\langle \frac{\partial f_\theta(\boldsymbol{x})}{\partial \boldsymbol{v}}, \frac{\partial f_\theta(\boldsymbol{x}')}{\partial \boldsymbol{v}} \right\rangle = \left(\frac{\sigma_v^2}{n}\left\langle \boldsymbol{h}^{(T)}(\boldsymbol{x}), \boldsymbol{h}^{(T')}(\boldsymbol{x}')\right\rangle\right).$$

Remember that for any two $G$-var $\mathbb{E}\left[[\boldsymbol{g}]_i[\boldsymbol{g}']_i\right]$ is independent of index $i$. Therefore,

$$\frac{1}{n}\left\langle \boldsymbol{h}^{(t-1)}(\boldsymbol{x}), \boldsymbol{h}^{(t'-1)}(\boldsymbol{x}')\right\rangle \to \mathrm{V}_\phi\left[\boldsymbol{K}^{(t,t')}(\boldsymbol{x}, \boldsymbol{x}')\right] \qquad t > 1$$

$$\frac{1}{n}\left\langle \boldsymbol{h}^{(0)}(\boldsymbol{x}), \boldsymbol{h}^{(0)}(\boldsymbol{x}')\right\rangle \to \sigma_h^2.$$

Hence, by summing the above terms in the infinite-width limit we get

$$\langle \nabla_\theta f_\theta(\boldsymbol{x}), \nabla_\theta f_\theta(\boldsymbol{x}')\rangle \to \left(\sum_{t'=1}^{T'}\sum_{t=1}^{T} \Pi^{(t,t')}(\boldsymbol{x}, \boldsymbol{x}') \cdot \Sigma^{(t,t')}(\boldsymbol{x}', \boldsymbol{x}')\right) + \mathcal{K}(\boldsymbol{x}, \boldsymbol{x}'). \qquad (32)$$

Since $\Pi^{(t,t')}(\boldsymbol{x}, \boldsymbol{x}') = 0$ for $t' - t \neq \tau$ it is simplified to

$$\langle \nabla_\theta f_\theta(\boldsymbol{x}), \nabla_\theta f_\theta(\boldsymbol{x}')\rangle = \left(\sum_{t=1}^{T} \Pi^{(t,t+\tau)}(\boldsymbol{x}, \boldsymbol{x}') \cdot \Sigma^{(t,t+\tau)}(\boldsymbol{x}', \boldsymbol{x}')\right) + \mathcal{K}(\boldsymbol{x}, \boldsymbol{x}').$$

**Multi-dimensional output.** For $f_\theta(\boldsymbol{x}) \in \mathbb{R}^d$, the $i$-th output for $i \in [d]$ is obtained via

$$[f_\theta(\boldsymbol{x})]_i = \frac{\sigma_v}{\sqrt{n}}\boldsymbol{v}_i^\top \boldsymbol{h}^{(T)}(\boldsymbol{x}),$$

where $\boldsymbol{v}_i$ is independent of $\boldsymbol{v}_j$ for $i \neq j$. As a result, for The RNTK $\Theta(\boldsymbol{x}, \boldsymbol{x}') \in \mathbb{R}^{d\times d}$ for multi-dimensional output we have

$$[\Theta(\boldsymbol{x}, \boldsymbol{x}')]_{i,j} = \left\langle \nabla_\theta\left[f_\theta(\boldsymbol{x})\right]_i, \nabla_\theta\left[f_\theta(\boldsymbol{x}')\right]_j\right\rangle$$

For $i = j$, the kernel is the same as computed in (32) and we denote it as

$$\langle \nabla_\theta\left[f_\theta(\boldsymbol{x})\right]_i, \nabla_\theta\left[f_\theta(\boldsymbol{x}')\right]_i\rangle = \Theta^{(T,T')}(\boldsymbol{x}, \boldsymbol{x}').$$

For $i \neq j$, since $\boldsymbol{v}_i$ is independent of $\boldsymbol{v}_j$, $\Pi^{(T,T')}(\boldsymbol{x}, \boldsymbol{x}')$ and all the backward pass gradients become zero, so

$$\left\langle \nabla_\theta \left[ f_\theta(\boldsymbol{x}) \right]_i, \nabla_\theta \left[ f_\theta(\boldsymbol{x}') \right]_j \right\rangle = 0 \qquad i \neq j$$

which gives us the following formula

$$\Theta(\boldsymbol{x}, \boldsymbol{x}') = \Theta^{(T,T')}(\boldsymbol{x}, \boldsymbol{x}') \otimes \boldsymbol{I}_d.$$

This concludes the proof for Theorem 1 for single-layer case.

## B.3 PROOF FOR THEOREM 1: MULTI-LAYER CASE

Now we drive the RNTK for multi-layer RNTK. We will only study single output case and the generalization to multi-dimensional case is identical as the single layer case. The set of equations for calculation of the output of a $L$-layer RNN for $\boldsymbol{x} = \{\boldsymbol{x}_t\}_{t=1}^T$ are

$$\boldsymbol{g}^{(\ell,t)}(\boldsymbol{x}) = \frac{\sigma_w^\ell}{\sqrt{n}} \mathbf{W}^{(\ell)} \boldsymbol{h}^{(\ell,t-1)}(\boldsymbol{x}) + \frac{\sigma_u^\ell}{\sqrt{m}} \mathbf{U}^{(\ell)} \boldsymbol{x}_t + \sigma_b^\ell \mathbf{b}^{(\ell)} \qquad \ell = 1$$

$$\boldsymbol{g}^{(\ell,t)}(\boldsymbol{x}) = \frac{\sigma_w^\ell}{\sqrt{n}} \mathbf{W}^{(\ell)} \boldsymbol{h}^{(\ell,t-1)}(\boldsymbol{x}) + \frac{\sigma_u^\ell}{\sqrt{n}} \mathbf{U}^{(\ell)} \boldsymbol{h}^{(\ell-1,t)}(\boldsymbol{x}) + \sigma_b^\ell \mathbf{b}^{(\ell)} \qquad \ell > 1$$

$$\boldsymbol{h}^{(\ell,t)}(\boldsymbol{x}) = \phi \left( \boldsymbol{g}^{(\ell,t)}(\boldsymbol{x}) \right)$$

$$f_\theta(\boldsymbol{x}) = \frac{\sigma_v}{\sqrt{n}} \boldsymbol{v}^\top \boldsymbol{h}^{(L,T)}(\boldsymbol{x})$$

The forward pass kernels for the first layer is the same as calculated in B.2. For $\ell \geq 2$ we have:

$$\Sigma^{(\ell,t,t')}(\boldsymbol{x}, \boldsymbol{x}') = \Sigma(\boldsymbol{g}^{(\ell,t)}(\boldsymbol{x}), \boldsymbol{g}^{(\ell,t')}(\boldsymbol{x}'))$$

$$= \Sigma \left( \frac{\sigma_w^\ell}{\sqrt{n}} \mathbf{W}^{(\ell)} \boldsymbol{h}^{(\ell,t-1)}(\boldsymbol{x}), \frac{\sigma_w^\ell}{\sqrt{n}} \mathbf{W}^{(\ell)} \boldsymbol{h}^{(\ell,t'-1)}(\boldsymbol{x}') \right)$$

$$+ \Sigma \left( \frac{\sigma_u^\ell}{\sqrt{n}} \mathbf{U}^{(\ell)} \boldsymbol{h}^{(\ell-1,t)}(\boldsymbol{x}), \frac{\sigma_u^\ell}{\sqrt{n}} \mathbf{U}^{(\ell)} \boldsymbol{h}^{(\ell-1,t')}(\boldsymbol{x}') \right) + \Sigma^{\mathrm{in}} \left( \sigma_b^\ell \boldsymbol{b}^{(\ell)}, \sigma_b^\ell \boldsymbol{b}^{(\ell)} \right)$$

$$= (\sigma_w^\ell)^2 \mathrm{V}_\phi \left[ \boldsymbol{K}^{(\ell,t,t')}(\boldsymbol{x}, \boldsymbol{x}') \right] + (\sigma_u^\ell)^2 \mathrm{V}_\phi \left[ \boldsymbol{K}^{(\ell-1,t+1,t'+1)}(\boldsymbol{x}, \boldsymbol{x}') \right] + (\sigma_b^\ell)^2,$$

where

$$\boldsymbol{K}^{(\ell,t,t')}(\boldsymbol{x}, \boldsymbol{x}') = \left[ \begin{array}{cc} \boldsymbol{\Sigma}^{(\ell,t-1,t-1)}(\boldsymbol{x}, \boldsymbol{x}) & \boldsymbol{\Sigma}^{(\ell,t-1,t'-1)}(\boldsymbol{x}, \boldsymbol{x}') \\ \boldsymbol{\Sigma}^{(\ell,t-1,t'-1)}(\boldsymbol{x}, \boldsymbol{x}') & \boldsymbol{\Sigma}^{(\ell,t'-1,t'-1)}(\boldsymbol{x}', \boldsymbol{x}') \end{array} \right],$$

and $\Sigma^{\mathrm{in}}$ is defined in (B.2). For the first first time step we have:

$$\Sigma^{(\ell,1,1)}(\boldsymbol{x}, \boldsymbol{x}') = (\sigma_w^\ell)^2 \sigma_h^2 1_{(\boldsymbol{x}=\boldsymbol{x}')} + (\sigma_u^\ell)^2 \mathrm{V}_\phi \left[ \boldsymbol{K}^{(\ell,2,2)}(\boldsymbol{x}, \boldsymbol{x}') \right] + (\sigma_b^\ell)^2,$$

$$\Sigma^{(\ell,t,1)}(\boldsymbol{x}, \boldsymbol{x}') = (\sigma_u^\ell)^2 \mathrm{V}_\phi \left[ \boldsymbol{K}^{(\ell,t+1,2)}(\boldsymbol{x}, \boldsymbol{x}') \right] + (\sigma_b^\ell)^2,$$

$$\Sigma^{(\ell,1,t')}(\boldsymbol{x}, \boldsymbol{x}') = (\sigma_u^\ell)^2 \mathrm{V}_\phi \left[ \boldsymbol{K}^{(\ell,2,t'+1)}(\boldsymbol{x}, \boldsymbol{x}') \right] + (\sigma_b^\ell)^2.$$

And the output layer

$$\mathcal{K}(\boldsymbol{x}, \boldsymbol{x}') = \sigma_v^2 \mathrm{V}_\phi \left[ \boldsymbol{K}^{(L,T+1,T'+1)}(\boldsymbol{x}, \boldsymbol{x}') \right].$$

Note that because of using new weights at each layer we get

$$\Sigma(\boldsymbol{g}^{(\ell,t)}(\boldsymbol{x}), \boldsymbol{g}^{(\ell',t')})(\boldsymbol{x})) = 0 \qquad \ell \neq \ell'$$

Now we calculate the backward pass kernels in multi-layer RNTK. The gradients at the last layer is calculated via

$$\boldsymbol{\delta}^{(L,T)}(\boldsymbol{x}) = \sigma_v \boldsymbol{v} \odot \phi'(\boldsymbol{g}^{(L,T)}(\boldsymbol{x})).$$

In the last hidden layer for different time steps we have

$$\boldsymbol{\delta}^{(L,t)}(\boldsymbol{x}) = \frac{\sigma_w^L}{\sqrt{n}} \left( \mathbf{W}^{(L)} \right)^\top \left( \phi'(\boldsymbol{g}^{(L,t)}(\boldsymbol{x})) \odot \boldsymbol{\delta}^{(L,t+1)}(\boldsymbol{x}) \right) \qquad t \in [T-1]$$

In the last time step for different hidden layers we have

$$\boldsymbol{\delta}^{(\ell,T)}(\boldsymbol{x}) = \frac{\sigma_u^{\ell+1}}{\sqrt{n}} \left(\mathbf{U}^{(\ell+1)}\right)^\top \left(\phi'(\boldsymbol{g}^{(\ell,T)}(\boldsymbol{x})) \odot \boldsymbol{\delta}^{(\ell+1,T)}(\boldsymbol{x})\right) \qquad \ell \in [L-1]$$

At the end for the other layers we have

$$\boldsymbol{\delta}^{(\ell,t)}(\boldsymbol{x}) = \frac{\sigma_w^\ell}{\sqrt{n}} \left(\mathbf{W}^{(\ell)}\right)^\top \left(\phi'(\boldsymbol{g}^{(\ell,t)}(\boldsymbol{x})) \odot \boldsymbol{\delta}^{(\ell,t+1)}(\boldsymbol{x})\right)$$
$$+ \frac{\sigma_u^{\ell+1}}{\sqrt{n}} \left(\mathbf{U}^{(\ell+1)}\right)^\top \left(\phi'(\boldsymbol{g}^{(\ell,t)}(\boldsymbol{x})) \odot \boldsymbol{\delta}^{(\ell+1,t)}(\boldsymbol{x})\right) \qquad \ell \in [L-1], t \in [T-1]$$

The recursive formula for the $\Pi^{(L,t,t')}(\boldsymbol{x},\boldsymbol{x}')$ is the same as the single layer, and it is non-zero for $t' - t = T' - T = \tau$. As a result we have

$$\Pi^{(L,T,T+\tau)}(\boldsymbol{x},\boldsymbol{x}') = \sigma_v^2 \mathrm{V}_{\phi'}\left[\boldsymbol{K}^{(L,T+1,T+\tau+1)}\right](\boldsymbol{x},\boldsymbol{x}')$$
$$\Pi^{(L,t,t+\tau)}(\boldsymbol{x},\boldsymbol{x}') = (\sigma_w^L)^2 \mathrm{V}_{\phi'}\left[\boldsymbol{K}^{(L,t+1,t+\tau+1)}\right](\boldsymbol{x},\boldsymbol{x}') \cdot \Pi^{(L,t+1,t+1+\tau)}(\boldsymbol{x},\boldsymbol{x}') \quad t \in [T-1]$$
$$\Pi^{(L,t,t')}(\boldsymbol{x},\boldsymbol{x}') = 0 \qquad\qquad\qquad\qquad\qquad\qquad\qquad\qquad\qquad\qquad t' - t \neq \tau \tag{33}$$

Similarly by using the same course of arguments used in the single layer setting, for the last time step we have

$$\Pi^{(\ell,T,T+\tau)}(\boldsymbol{x},\boldsymbol{x}') = (\sigma_u^{\ell+1})^2 \mathrm{V}_{\phi'}\left[\boldsymbol{K}^{(\ell,T+1,T+\tau+1)}\right](\boldsymbol{x},\boldsymbol{x}') \cdot \Pi^{(\ell+1,T+1,T+\tau+1)}(\boldsymbol{x},\boldsymbol{x}') \qquad \ell \in [L-1]$$

For the other layers we have

$$\Pi^{(\ell,t,t')}(\boldsymbol{x},\boldsymbol{x}') = (\sigma_w^\ell)^2 \mathrm{V}_{\phi'}\left[\boldsymbol{K}^{(\ell,t+1,t+\tau+1)}\right](\boldsymbol{x},\boldsymbol{x}') \cdot \Pi^{(\ell,t+1,t+1+\tau)}(\boldsymbol{x},\boldsymbol{x}')$$
$$+ (\sigma_u^{\ell+1})^2 \mathrm{V}_{\phi'}\left[\boldsymbol{K}^{(\ell,t+1,t+\tau+1)}\right](\boldsymbol{x},\boldsymbol{x}') \cdot \Pi^{(\ell+1,t+1,t'+1)}(\boldsymbol{x},\boldsymbol{x}').$$

For $t' - t \neq \tau$ the recursion continues until it reaches $\Pi^{(L,T,t'')}(\boldsymbol{x},\boldsymbol{x}'), t'' < T'$ or $\Pi^{(L,t'',T')}(\boldsymbol{x},\boldsymbol{x}'), t'' < T$ and as a result based on (33) we get

$$\Pi^{(\ell,t,t')}(\boldsymbol{x},\boldsymbol{x}') = 0 \qquad t' - t \neq \tau \tag{34}$$

For $t' - t = \tau$ it leads to $\Pi^{(L,T,T')}(\boldsymbol{x},\boldsymbol{x}')$ and has a non-zero value.

Now we derive RNTK for multi-layer:

$$\langle \nabla_\theta f_\theta(\boldsymbol{x}), \nabla_\theta f_\theta(\boldsymbol{x}') \rangle = \sum_{\ell=1}^L \left\langle \frac{\partial f_\theta(\boldsymbol{x})}{\partial \mathbf{W}^{(\ell)}}, \frac{\partial f_\theta(\boldsymbol{x}')}{\partial \mathbf{W}^{(\ell)}} \right\rangle + \sum_{\ell=1}^L \left\langle \frac{\partial f_\theta(\boldsymbol{x})}{\partial \mathbf{U}^{(\ell)}}, \frac{\partial f_\theta(\boldsymbol{x}')}{\partial \mathbf{U}^{(\ell)}} \right\rangle$$
$$+ \sum_{\ell=1}^L \left\langle \frac{\partial f_\theta(\boldsymbol{x})}{\partial \mathbf{b}^{(\ell)}}, \frac{\partial f_\theta(\boldsymbol{x}')}{\partial \mathbf{b}^{(\ell)}} \right\rangle + \left\langle \frac{\partial f_\theta(\boldsymbol{x})}{\partial \boldsymbol{v}}, \frac{\partial f_\theta(\boldsymbol{x}')}{\partial \boldsymbol{v}} \right\rangle,$$

where

$$\left\langle \frac{\partial f_\theta(\boldsymbol{x})}{\partial \mathbf{W}^{(\ell)}}, \frac{\partial f_\theta(\boldsymbol{x}')}{\partial \mathbf{W}^{(\ell)}} \right\rangle = \sum_{t'=1}^{T'} \sum_{t=1}^T \left(\frac{1}{n} \left\langle \boldsymbol{\delta}^{(\ell,t)}(\boldsymbol{x}), \boldsymbol{\delta}^{(\ell,t')}(\boldsymbol{x}') \right\rangle\right) \cdot \left(\frac{(\sigma_w^\ell)^2}{n} \left\langle \boldsymbol{h}^{(\ell,t-1)}(\boldsymbol{x}), \boldsymbol{h}^{(\ell,t'-1)}(\boldsymbol{x}') \right\rangle\right)$$

$$\left\langle \frac{\partial f_\theta(\boldsymbol{x})}{\partial \mathbf{U}^{(\ell)}}, \frac{\partial f_\theta(\boldsymbol{x}')}{\partial \mathbf{U}^{(\ell)}} \right\rangle = \sum_{t'=1}^{T'} \sum_{t=1}^T \left(\frac{1}{n} \left\langle \boldsymbol{\delta}^{(\ell,t)}(\boldsymbol{x}), \boldsymbol{\delta}^{(\ell,t')}(\boldsymbol{x}') \right\rangle\right) \cdot \left(\frac{(\sigma_u^\ell)^2}{m} \langle \boldsymbol{x}_t, \boldsymbol{x}'_{t'} \rangle\right) \qquad \ell = 1$$

$$\left\langle \frac{\partial f_\theta(\boldsymbol{x})}{\partial \mathbf{U}^{(\ell)}}, \frac{\partial f_\theta(\boldsymbol{x}')}{\partial \mathbf{U}^{(\ell)}} \right\rangle = \sum_{t'=1}^{T'} \sum_{t=1}^T \left[\left(\frac{1}{n} \left\langle \boldsymbol{\delta}^{(\ell,t)}(\boldsymbol{x}), \boldsymbol{\delta}^{(\ell,t')}(\boldsymbol{x}') \right\rangle\right)\right.$$
$$\left. \cdot \left(\frac{(\sigma_u^\ell)^2}{n} \left\langle \boldsymbol{h}^{(\ell-1,t)}(\boldsymbol{x}), \boldsymbol{h}^{(\ell-1,t')}(\boldsymbol{x}') \right\rangle\right)\right] \qquad \ell > 1$$

$$\left\langle \frac{\partial f_\theta(\boldsymbol{x})}{\partial \mathbf{b}^{(\ell)}}, \frac{\partial f_\theta(\boldsymbol{x}')}{\partial \mathbf{b}^{(\ell)}} \right\rangle = \sum_{t'=1}^{T'} \sum_{t=1}^T \left(\frac{1}{n} \left\langle \boldsymbol{\delta}^{(\ell,t)}(\boldsymbol{x}), \boldsymbol{\delta}^{(\ell,t')}(\boldsymbol{x}') \right\rangle\right) \cdot (\sigma_b^\ell)^2$$

$$\left\langle \frac{\partial f_\theta(\boldsymbol{x})}{\partial \boldsymbol{v}}, \frac{\partial f_\theta(\boldsymbol{x}')}{\partial \boldsymbol{v}} \right\rangle = \left(\frac{\sigma_v^2}{n} \left\langle \boldsymbol{h}^{(T)}(\boldsymbol{x}), \boldsymbol{h}^{(T')}(\boldsymbol{x}') \right\rangle\right)$$

Summing up all the terms and replacing the inner product of vectors with their expectations we get

$$\langle \nabla_\theta f_\theta(\boldsymbol{x}), \nabla_\theta f_\theta(\boldsymbol{x}')\rangle = \Theta^{(L,T,T')} = \left( \sum_{\ell=1}^{L} \sum_{t=1}^{T} \sum_{t'=1}^{T'} \Pi^{(\ell,t,t')}(\boldsymbol{x},\boldsymbol{x}') \cdot \Sigma^{(\ell,t,t')}(\boldsymbol{x},\boldsymbol{x}') \right) + \mathcal{K}(\boldsymbol{x},\boldsymbol{x}').$$

By (34), we can simplify to

$$\Theta^{(L,T,T')} = \left( \sum_{\ell=1}^{L} \sum_{t=1}^{T} \Pi^{(\ell,t,t')}(\boldsymbol{x},\boldsymbol{x}') \cdot \Sigma^{(\ell,t,t+\tau)}(\boldsymbol{x},\boldsymbol{x}') \right) + \mathcal{K}(\boldsymbol{x},\boldsymbol{x}').$$

For multi-dimensional output it becomes

$$\Theta(\boldsymbol{x},\boldsymbol{x}') = \Theta^{(L,T,T')}(\boldsymbol{x},\boldsymbol{x}') \otimes \boldsymbol{I}_d.$$

This concludes the proof for Theorem 1 for the multi-layer case.

### B.4 PROOF FOR THEOREM 3: WEIGHT-UNTIED RNTK

The architecture of a weight-untied single layer RNN is

$$\boldsymbol{g}^{(t)}(\boldsymbol{x}) = \frac{\sigma_w}{\sqrt{m}} \mathbf{W}^{(t)} \boldsymbol{h}^{(t-1)}(\boldsymbol{x}) + \frac{\sigma_u}{\sqrt{n}} \mathbf{U}^{(t)} \boldsymbol{x}_t + \sigma_b \mathbf{b}^{(t)}$$

$$\boldsymbol{h}^{(t)}(\boldsymbol{x}) = \phi\left(\boldsymbol{g}^{(t)}(\boldsymbol{x})\right)$$

$$f_\theta(\boldsymbol{x}) = \frac{\sigma_v}{\sqrt{n}} \boldsymbol{v}^\top \boldsymbol{h}^{(T)}(\boldsymbol{x})$$

Where we use new weights at each time step and we index it by time. Like previous sections, we first derive the forward pass kernels for two same length data $\boldsymbol{x} = \{\boldsymbol{x}_t\}_{t=1}^{T}, \boldsymbol{x} = \{\boldsymbol{x}_{t'}'\}_{t'=1}^{T}$

$$\Sigma^{(t,t)}(\boldsymbol{x},\boldsymbol{x}') = \sigma_w^2 \mathrm{V}_\phi\left[\boldsymbol{K}^{(t,t)}(\boldsymbol{x},\boldsymbol{x}')\right] + \frac{\sigma_u^2}{m}\langle \boldsymbol{x}_t, \boldsymbol{x}_t'\rangle + \sigma_b^2.$$

$$\Sigma^{(t,t')}(\boldsymbol{x},\boldsymbol{x}') = 0 \qquad t \neq t'$$

Since we are using same weight at the same time step, $\Sigma^{(t,t)}(\boldsymbol{x},\boldsymbol{x}')$ can be written as a function of the previous kernel, which is exactly as the weight-tied RNN. However for different length, it becomes zero as a consequence of using different weights, unlike weight-tied which has non-zero value. The kernel of the first time step and output is also the same as weight-tied RNN. For the gradients we have:

$$\boldsymbol{\delta}^{(T)}(\boldsymbol{x}) = \sigma_v \boldsymbol{v} \odot \phi'(\boldsymbol{g}^{(T)}(\boldsymbol{x}))$$

$$\boldsymbol{\delta}^{(t)}(\boldsymbol{x}) = \frac{\sigma_w}{\sqrt{n}} (\mathbf{W}^{(t+1)})^\top \left(\phi'(\boldsymbol{g}^{(t)}(\boldsymbol{x})) \odot \boldsymbol{\delta}^{(t+1)}(\boldsymbol{x})\right) \qquad t \in [T-1]$$

For $t' = t$ we have:

$$\Pi^{(t,t)}(\boldsymbol{x},\boldsymbol{x}') = \sigma_w^2 \mathrm{V}_{\phi'}\left[\boldsymbol{K}^{(t+1,t+1)}(\boldsymbol{x},\boldsymbol{x}')\right]\Pi^{(t+1,t+1)}(\boldsymbol{x},\boldsymbol{x}')$$

$$\Pi^{(t,t)}(\boldsymbol{x},\boldsymbol{x}') = \sigma_v^2 \mathrm{V}_{\phi'}\left[\boldsymbol{K}^{(T+1,T+\tau+1)}(\boldsymbol{x},\boldsymbol{x}')\right].$$

Due to using different weights for $t \neq t'$, we can immediately conclude that $\Pi^{(t,t')}(\boldsymbol{x},\boldsymbol{x}') = 0$. This set of calculation is exactly the same as the weight-tied case when $\tau = T - T = 0$.

Finally, with $\theta = \mathrm{Vect}\left[\{\{\mathbf{W}^{(t)}, \mathbf{U}^{(t)}, \mathbf{b}^{(t)}\}_{t=1}^{T}, \boldsymbol{v}\}\right]$ we have

$$\langle \nabla_\theta f_\theta(\boldsymbol{x}), \nabla_\theta f_\theta(\boldsymbol{x}')\rangle = \sum_{t=1}^{T} \left\langle \frac{\partial f_\theta(\boldsymbol{x})}{\partial \mathbf{W}^{(t)}}, \frac{\partial f_\theta(\boldsymbol{x}')}{\partial \mathbf{W}^{(t)}} \right\rangle + \sum_{t=1}^{T} \left\langle \frac{\partial f_\theta(\boldsymbol{x})}{\partial \mathbf{U}^{(t)}}, \frac{\partial f_\theta(\boldsymbol{x}')}{\partial \mathbf{U}^{(t)}} \right\rangle$$

$$+ \sum_{t=1}^{T} \left\langle \frac{\partial f_\theta(\boldsymbol{x})}{\partial \mathbf{b}^{(t)}}, \frac{\partial f_\theta(\boldsymbol{x}')}{\partial \mathbf{b}^{(t)}} \right\rangle + \left\langle \frac{\partial f_\theta(\boldsymbol{x})}{\partial \boldsymbol{v}}, \frac{\partial f_\theta(\boldsymbol{x}')}{\partial \boldsymbol{v}}, \right\rangle$$

with

$$\left\langle \frac{\partial f_\theta(\boldsymbol{x})}{\partial \mathbf{W}^{(t)}}, \frac{\partial f_\theta(\boldsymbol{x}')}{\partial \mathbf{W}^{(t)}} \right\rangle = \left( \frac{1}{n} \left\langle \boldsymbol{\delta}^{(t)}(\boldsymbol{x}), \boldsymbol{\delta}^{(t)}(\boldsymbol{x}') \right\rangle \right) \cdot \left( \frac{\sigma_w^2}{n} \left\langle \boldsymbol{h}^{(t-1)}(\boldsymbol{x}), \boldsymbol{h}^{(t-1)}(\boldsymbol{x}') \right\rangle \right)$$

$$\left\langle \frac{\partial f_\theta(\boldsymbol{x})}{\partial \mathbf{U}^{(t)}}, \frac{\partial f_\theta(\boldsymbol{x}')}{\partial \mathbf{U}^{(t)}} \right\rangle = \left( \frac{1}{n} \left\langle \boldsymbol{\delta}^{(t)}(\boldsymbol{x}), \boldsymbol{\delta}^{(t)}(\boldsymbol{x}') \right\rangle \right) \cdot \left( \frac{\sigma_u^2}{m} \left\langle \boldsymbol{x}_t, \boldsymbol{x}'_t \right\rangle \right)$$

$$\left\langle \frac{\partial f_\theta(\boldsymbol{x})}{\partial \mathbf{b}^{(t)}}, \frac{\partial f_\theta(\boldsymbol{x}')}{\partial \mathbf{b}^{(t)}} \right\rangle = \left( \frac{1}{n} \left\langle \boldsymbol{\delta}^{(t)}(\boldsymbol{x}), \boldsymbol{\delta}^{(t)}(\boldsymbol{x}') \right\rangle \right) \cdot \sigma_b^2$$

$$\left\langle \frac{\partial f_\theta(\boldsymbol{x})}{\partial \boldsymbol{v}}, \frac{\partial f_\theta(\boldsymbol{x}')}{\partial \boldsymbol{v}} \right\rangle = \left( \frac{\sigma_v^2}{n} \left\langle \boldsymbol{h}^{(T)}(\boldsymbol{x}), \boldsymbol{h}^{(T')}(\boldsymbol{x}') \right\rangle \right).$$

As a result we obtain

$$\langle \nabla_\theta f_\theta(\boldsymbol{x}), \nabla_\theta f_\theta(\boldsymbol{x}') \rangle = \left( \sum_{t=1}^{T} \Pi^{(t,t)}(\boldsymbol{x}, \boldsymbol{x}') \cdot \Sigma^{(t,t)}(\boldsymbol{x}', \boldsymbol{x}') \right) + \mathcal{K}(\boldsymbol{x}, \boldsymbol{x}'),$$

same as the weight-tied RNN when $\tau = 0$. This concludes the proof for Theorem 3.

### B.5 ANALYTICAL FORMULA FOR $V_\phi[K]$

For any positive definite matrix $\boldsymbol{K} = \begin{bmatrix} K_1 & K_3 \\ K_3 & K_2 \end{bmatrix}$ we have:

- $\phi = \text{ReLU}$ (Cho & Saul, 2009)

$$V_\phi[\boldsymbol{K}] = \frac{1}{2\pi} \left( c(\pi - \arccos(c)) + \sqrt{1 - c^2} \right) \sqrt{K_1 K_2},$$

$$V_{\phi'}[\boldsymbol{K}] = \frac{1}{2\pi} (\pi - \arccos(c)).$$

where $c = K_3/\sqrt{K_1 K_2}$

- $\phi = \text{erf}$ (Neal, 1995)

$$V_\phi[\boldsymbol{K}] = \frac{2}{\pi} \arcsin \left( \frac{2K_3}{\sqrt{(1 + 2K_1)(1 + 2K_3)}} \right),$$

$$V_{\phi'}[\boldsymbol{K}] = \frac{4}{\pi \sqrt{(1 + 2K_1)(1 + 2K_2) - 4K_3^2}}.$$

## C  PROOF FOR THEOREM 2: RNTK CONVERGENCE AFTER TRAINING

To prove theorem 2, we use the strategy used in (Lee et al., 2019) which relies on the the local lipschitzness of the network Jacobian $\boldsymbol{J}(\theta, \mathcal{X}) = \nabla_\theta f_\theta(\boldsymbol{x}) \in \mathbb{R}^{|\mathcal{X}|d \times |\theta|}$ at initialization.

**Definition 1** *The Jacobian of a neural network is local lipschitz at NTK initialization $(\theta_0 \sim \mathcal{N}(0,1))$ if there is constant $K > 0$ for every $C$ such that*

$$\begin{cases} \|\boldsymbol{J}(\theta, \mathcal{X})\|_F < K \\ \|\boldsymbol{J}(\theta, \mathcal{X}) - \boldsymbol{J}(\tilde{\theta}, \mathcal{X})\|_F < K\|\theta - \tilde{\theta}\| \end{cases}, \qquad \forall\, \theta, \tilde{\theta} \in B(\theta_0, R)$$

*where*

$$B(\theta, R) := \{\theta : \|\theta_0 - \theta\| < R\}.$$

**Theorem 4** *Assume that the network Jacobian is local lipschitz with high probability and the empirical NTK of the network converges in probability at initialization and it is positive definite over the input set. For $\epsilon > 0$, there exists $N$ such that for $n > N$ when applying gradient flow with $\eta < 2 \left( \lambda_{\min}(\Theta(\mathcal{X}, \mathcal{X})) + \lambda_{\max}(\Theta(\mathcal{X}, \mathcal{X}))\right)^{-1}$ with probability at least $(1 - \epsilon)$ we have:*

$$\sup_s \frac{\|\theta_s - \theta_0\|_2}{\sqrt{n}}, \sup_s \|\widehat{\Theta}_s(\mathcal{X}, \mathcal{X}) - \widehat{\Theta}_0(\mathcal{X}, \mathcal{X})\| = \mathcal{O}\left( \frac{1}{\sqrt{n}} \right).$$

*Proof: See (Lee et al., 2019)*

Theorem 4 holds for any network architecture and any cost function and it was used in (Lee et al., 2019) to show the stability of NTK for MLP during training.

Here we extend the results for RNTK by proving that the Jacobian of a multi-layer RNN under NTK initialization is local lipschitz with high probability.

To prove it, first, we prove that for any two points $\theta, \tilde{\theta} \in B(\theta_0, R)$ there exists constant $K_1$ such that

$$\|\boldsymbol{g}^{(\ell,t)}(\boldsymbol{x})\|_2, \|\delta^{(\ell,t)}(\boldsymbol{x})\|_2 \leq K_1 \sqrt{n} \tag{35}$$

$$\|\boldsymbol{g}^{(\ell,t)}(\boldsymbol{x}) - \tilde{\boldsymbol{g}}^{(\ell,t)}(\boldsymbol{x})\|_2, \|\boldsymbol{\delta}^{(\ell,t)}(\boldsymbol{x}) - \tilde{\boldsymbol{\delta}}^{(\ell,t)}(\boldsymbol{x})\|_2 \leq \|\bar{\theta} - \tilde{\theta}\| \leq K_1 \sqrt{n}\|\theta - \tilde{\theta}\|. \tag{36}$$

To prove (35) and (36) we use the following lemmas.[3]

**Lemma 1** *Let $A \in \mathbb{R}^{n \times m}$ be a random matrix whose entries are independent standard normal random variables. Then for every $t \geq 0$, with probability at least $1 - e^{(-ct^2)}$ for some constant $c$ we have:*

$$\|A\|_2 \leq \sqrt{m} + \sqrt{n} + t.$$

**Lemma 2** *Let $a \in \mathbb{R}^n$ be a random vector whose entries are independent standard normal random variables. Then for every $t \geq 0$, with probability at least $1 - e^{(-ct^2)}$ for some constant $c$ we have:*

$$\|a\|_2 \leq \sqrt{n} + \sqrt{t}.$$

Setting $t = \sqrt{n}$ for any $\theta \in R(\theta_0, R)$. With high probability, we get:

$$\|\mathbf{W}^{(\ell)}\|_2, \|\mathbf{U}^{(\ell)}\|_2 \leq 3\sqrt{n}, \quad \|\mathbf{b}^\ell\|_2 \leq 2\sqrt{n}, \quad \|\mathbf{h}^{(\ell,0)}(\boldsymbol{x})\|_2 \leq 2\sigma_h\sqrt{n}.$$

We also assume that there exists some finite constant $C$ such that

$$|\phi(x)| < C|x|, \quad |\phi(x) - \phi(x')| < C|x - x'|, \quad |\phi'(x)| < C, \quad , |\phi'(x) - \phi'(x')| < C|x - x'|.$$

The proof is obtained by induction. From now on assume that all inequalities in (35) and (36) holds with some $k$ for the previous layers. We have

$$\|\boldsymbol{g}^{(\ell,t)}(\boldsymbol{x})\|_2 = \|\frac{\sigma_w^\ell}{\sqrt{n}}\mathbf{W}^{(\ell)}\boldsymbol{h}^{(\ell,t-1)}(\boldsymbol{x}) + \frac{\sigma_u^\ell}{\sqrt{n}}\mathbf{U}^{(\ell)}\boldsymbol{h}^{(\ell-1,t)}(\boldsymbol{x}) + \sigma_b^\ell\mathbf{b}^{(\ell)}\|_2$$

$$\leq \frac{\sigma_w^\ell}{\sqrt{n}}\|\mathbf{W}^{(\ell)}\|_2\|\phi\left(\boldsymbol{g}^{(\ell,t-1)}(\boldsymbol{x})\right)\|_2 + \frac{\sigma_u^\ell}{\sqrt{n}}\|\mathbf{U}^{(\ell)}\|_2\|\phi\left(\boldsymbol{g}^{(\ell-1,t)}(\boldsymbol{x})\right)\|_2 + \sigma_b^\ell\|\mathbf{b}^{(\ell)}\|_2$$

$$\leq \left(3\sigma_w^\ell Ck + 3\sigma_u^\ell Ck + 2\sigma_b\right)\sqrt{n}.$$

And the proof for (35) and (36) is completed by showing that the first layer is bounded

$$\|\boldsymbol{g}^{(1,1)}(\boldsymbol{x})\|_2 = \|\frac{\sigma_w^1}{\sqrt{n}}\mathbf{W}^{(\ell)}\boldsymbol{h}^{(1,0)}(\boldsymbol{x}) + \frac{\sigma_u^1}{\sqrt{m}}\mathbf{U}^{(\ell)}\boldsymbol{x}_1 + \sigma_b^1\mathbf{b}^{(1)}\|_2$$

$$\leq (3\sigma_w^1\sigma_h + \frac{3\sigma_u}{\sqrt{m}}\|\boldsymbol{x}_1\|_2 + 2\sigma_b)\sqrt{n}.$$

For the gradient of first layer we have

$$\|\boldsymbol{\delta}^{(L,T)}(\boldsymbol{x})\|_2 = \|\sigma_v\boldsymbol{v} \odot \phi'(\boldsymbol{g}^{(L,T)}(\boldsymbol{x}))\|_2$$

$$\leq \sigma_v\|\boldsymbol{v}\|_2\|\phi'(\boldsymbol{g}^{(L,T)}(\boldsymbol{x}))\|_\infty$$

$$= 2\sigma_v C\sqrt{n}.$$

And similarly we have

$$\|\boldsymbol{\delta}^{(\ell,t)}(\boldsymbol{x})\| \leq \left(3\sigma_w Ck' + 3\sigma_u Ck'\right)\sqrt{n}.$$

---

[3]See `math.uci.edu/~rvershyn/papers/HDP-book/HDP-book.pdf` for proofs

For $\theta, \tilde{\theta} \in B(\theta_0, R)$ we have

$$\|\boldsymbol{g}^{(1,1)}(\boldsymbol{x}) - \tilde{\boldsymbol{g}}^{(1,1)}(\boldsymbol{x})\|_2 = \|\frac{\sigma_w^1}{\sqrt{n}}(\mathbf{W}^{(1)} - \tilde{\mathbf{W}}^{(1)})\boldsymbol{h}^{(1,0)}(\boldsymbol{x}) + \frac{\sigma_u^1}{\sqrt{m}}(\mathbf{U}^{(1)} - \tilde{\mathbf{U}}^{(1)})\boldsymbol{h}^{(1,0)}(\boldsymbol{x})\|_2$$
$$\leq \left(3\sigma_w^1\sigma_h + \frac{3\sigma_u^1}{m}\|\boldsymbol{x}_1\|_2\right)\|\theta - \tilde{\theta}\|_2\sqrt{n}.$$

$$\|\boldsymbol{g}^{(\ell,t)}(\boldsymbol{x}) - \tilde{\boldsymbol{g}}^{(\ell,t)}(\boldsymbol{x})\|_2 \leq \|\phi(\boldsymbol{g}^{(\ell,t-1)}(\boldsymbol{x}))\|_2\|\frac{\sigma_w^\ell}{\sqrt{n}}(\mathbf{W}^{(\ell)} - \tilde{\mathbf{W}}^{(\ell)})\|_2$$
$$+ \|\frac{\sigma_w^\ell}{\sqrt{n}}\tilde{\mathbf{W}}^{(\ell)}\|_2\|\phi(\boldsymbol{g}^{(\ell,t-1)}(\boldsymbol{x})) - \phi(\tilde{\boldsymbol{g}}^{(\ell,t-1)}(\boldsymbol{x}))\|_2$$
$$+ \|\phi(\boldsymbol{g}^{(\ell-1,t)}(\boldsymbol{x}))\|_2\|\frac{\sigma_u^\ell}{\sqrt{n}}(\mathbf{U}^{(\ell)} - \tilde{\mathbf{U}}^{(\ell)})\|_2$$
$$+ \|\frac{\sigma_u^\ell}{\sqrt{n}}\tilde{\mathbf{U}}^{(\ell)}\|_2\|\phi(\boldsymbol{g}^{(\ell-1,t)}(\boldsymbol{x})) - \phi(\tilde{\boldsymbol{g}}^{(\ell-1,t)}(\boldsymbol{x}))\|_2 + \sigma_b\|\mathbf{b}^{(\ell)} - \tilde{\mathbf{b}}^{(\ell)}\|$$
$$\leq (k\sigma_w^\ell + 3\sigma_w^\ell Ck + k\sigma_u^\ell + 3\sigma_u^\ell Ck + \sigma_b)\|\theta - \tilde{\theta}\|_2\sqrt{n}.$$

For gradients we have

$$\|\boldsymbol{\delta}^{(L,T)}(\boldsymbol{x}) - \tilde{\boldsymbol{\delta}}^{(L,T)}(\boldsymbol{x})\|_2 \leq \sigma_v\|\phi'(\boldsymbol{g}^{(L,T)})\|_\infty\|(\boldsymbol{v} - \tilde{\boldsymbol{v}})\|_2 + \sigma_v\|\boldsymbol{v}\|_2\|\phi'(\boldsymbol{g}^{(L,T)}(\boldsymbol{x})) - \phi'(\boldsymbol{g}^{(L,T)}(\boldsymbol{x}))\|_2$$
$$\leq (\sigma_v C + 2\sigma_v Ck)\|\theta - \tilde{\theta}\|_2\sqrt{n}.$$

And similarly using same techniques we have

$$\|\boldsymbol{\delta}^{(\ell,t)}(\boldsymbol{x}) - \tilde{\boldsymbol{\delta}}^{(\ell,t)}(\boldsymbol{x})\|_2 \leq (\sigma_w C + 3\sigma_w Ck + \sigma_u C + 3\sigma_u Ck)\|\theta - \tilde{\theta}\|_2\sqrt{n}.$$

As a result, there exists $K_1$ that is a function of $\sigma_w, \sigma_u, \sigma_b, L, T$ and the norm of the inputs.

Now we prove the local Lipchitzness of the Jacobian

$$\|\boldsymbol{J}(\theta, \boldsymbol{x})\|_F \leq \sum_{\ell=2}^{L}\sum_{t=1}^{T}\left(\frac{1}{n}\left\|\boldsymbol{\delta}^{(\ell,t)}(\boldsymbol{x})\left(\sigma_w^\ell \boldsymbol{h}^{(\ell,t-1)}(\boldsymbol{x})\right)^\top\right\|_F\right.$$
$$+ \frac{1}{n}\left\|\boldsymbol{\delta}^{(\ell,t)}(\boldsymbol{x})\left(\sigma_u^\ell \boldsymbol{h}^{(\ell,t-1)}(\boldsymbol{x})\right)^\top\right\|_F + \frac{1}{\sqrt{n}}\left\|\boldsymbol{\delta}^{(\ell,t)}(\boldsymbol{x})\cdot\sigma_b^\ell\right\|_F\right)$$
$$+ \sum_{t=1}^{T}\left(\frac{1}{n}\left\|\boldsymbol{\delta}^{(1,t)}(\boldsymbol{x})\left(\sigma_w^1 \boldsymbol{h}^{(1,t-1)}(\boldsymbol{x})\right)^\top\right\|_F\right.$$
$$+ \frac{1}{\sqrt{nm}}\left\|\boldsymbol{\delta}^{(1,t-1)}(\boldsymbol{x})\left(\sigma_u^1 \boldsymbol{x}_t\right)^\top\right\|_F + \frac{1}{\sqrt{n}}\left\|\boldsymbol{\delta}^{(1,t)}(\boldsymbol{x})\cdot\sigma_b^1\right\|_F\right) + \frac{\sigma_v}{\sqrt{n}}\|\boldsymbol{h}^{(L,T)}(\boldsymbol{x})\|_F$$
$$\leq \left(\sum_{\ell=2}^{L}\sum_{t=1}^{T}(K_1^2 C\sigma_w^\ell + K_1^2 C\sigma_u^\ell + \sigma_b^\ell K_1)\right.$$
$$+ \sum_{t=1}^{T}(K_1^2 C\sigma_w^1 + \frac{K_1\sigma_u^1}{\sqrt{m}}\|\boldsymbol{x}_t\|_2 + \sigma_b^1 K_1) + \sigma_v C K_1\right).$$

And for $\theta, \tilde{\theta} \in B(\theta_0, R)$ we have

$$
\begin{aligned}
\| \boldsymbol{J}(\theta, \boldsymbol{x}) - \tilde{\boldsymbol{J}}(\theta, \boldsymbol{x}) \|_F \leq & \sum_{\ell=2}^{L} \sum_{t=1}^{T} \left( \frac{1}{n} \left\| \boldsymbol{\delta}^{(\ell,t)}(\boldsymbol{x}) \left( \sigma_w^\ell \boldsymbol{h}^{(\ell,t-1)}(\boldsymbol{x}) \right)^\top - \tilde{\boldsymbol{\delta}}^{(\ell,t)}(\boldsymbol{x}) \left( \sigma_w^\ell \tilde{\boldsymbol{h}}^{(\ell,t-1)}(\boldsymbol{x}) \right)^\top \right\|_F \right. \\
& + \frac{1}{n} \left\| \boldsymbol{\delta}^{(\ell,t)}(\boldsymbol{x}) \left( \sigma_u^\ell \boldsymbol{h}^{(\ell,t-1)}(\boldsymbol{x}) \right)^\top - \tilde{\boldsymbol{\delta}}^{(\ell,t)}(\boldsymbol{x}) \left( \sigma_u^\ell \tilde{\boldsymbol{h}}^{(\ell,t-1)}(\boldsymbol{x}) \right)^\top \right\|_F \\
& + \frac{1}{\sqrt{n}} \left\| \boldsymbol{\delta}^{(\ell,t)}(\boldsymbol{x}) \cdot \sigma_b^\ell - \tilde{\boldsymbol{\delta}}^{(\ell,t)}(\boldsymbol{x}) \cdot \sigma_b^\ell \right\|_F \\
& + \sum_{t=1}^{T} \left( \frac{1}{n} \left\| \boldsymbol{\delta}^{(1,t)}(\boldsymbol{x}) \left( \sigma_w^1 \boldsymbol{h}^{(1,t-1)}(\boldsymbol{x}) \right)^\top - \tilde{\boldsymbol{\delta}}^{(1,t)}(\boldsymbol{x}) \left( \sigma_w^1 \boldsymbol{h}^{(1,\tilde{t}-1)}(\boldsymbol{x}) \right)^\top \right\|_F \right. \\
& + \frac{1}{\sqrt{nm}} \left\| \boldsymbol{\delta}^{(1,t-1)}(\boldsymbol{x}) \left( \sigma_u^1 \boldsymbol{x}_t \right)^\top - \tilde{\boldsymbol{\delta}}^{(1,t-1)}(\boldsymbol{x}) \left( \sigma_u^1 \boldsymbol{x}_t \right)^\top \right\|_F \\
& + \left. \frac{1}{\sqrt{n}} \left\| \boldsymbol{\delta}^{(\ell,t)}(\boldsymbol{x}) \cdot \sigma_b^\ell - \tilde{\boldsymbol{\delta}}^{(\ell,t)}(\boldsymbol{x}) \cdot \sigma_b^\ell \right\| \right) + \frac{\sigma_v}{\sqrt{n}} \| \boldsymbol{h}^{(L,T)}(\boldsymbol{x}) - \boldsymbol{h}^{(L,\tilde{T})}(\boldsymbol{x}) \|_F \\
\leq & \left( \sum_{\ell=2}^{L} \sum_{t=1}^{T} (4 K_1^2 C \sigma_w^\ell + 4 K_1^2 C \sigma_u^\ell + \sigma_b^\ell K_1) \right. \\
& + \left. \sum_{t=1}^{T} (4 K_1^2 C \sigma_w^1 + \frac{K_1 \sigma_u^1}{\sqrt{m}} \| \boldsymbol{x}_t \|_2 + \sigma_b^1 K_1) + \sigma_v C K_1 \right) \| \theta - \tilde{\theta} \|_2 .
\end{aligned}
$$

The above proof can be generalized to the entire dataset by a straightforward application of the union bound. This concludes the proof for Theorem 2.

