# OpenReview forum: "The Recurrent Neural Tangent Kernel"
_ICLR.cc/2021/Conference — ICLR 2021 Poster_

### Official Review · AnonReviewer2 · 2020-10-27
**overall this is a good submission**

**Rating:** 6
**Confidence:** 4

**Review:**

This paper extends NTK to RNN to explain behavior of RNNs in overparametrized case. It’s a good extension study and interesting to see RNN with infinite-width limit converges to a kernel. The paper proves the same RNTK formula when the weights are shared and not shared. The proposed sensitivity for computationally friendly RNTK hyperparameter tuning is also insightful.

Weakness:

In the experimental part, the paper claims they restrict the data with shorter and fewer samples. This may be a downside of the proposed method as the goal of RNNs is to handle various length of data samples. Also, it seems that the proposed RNTK cannot outperform other SOA methods. Any reason? Why are GRU and identity RNNs chosen? Please highlight the best number for each experiment for an easier comparison.

---

> ### Author Response · Authors · 2020-11-15
> **Some clarifications regarding the experiments**
>
> We thank the reviewers for their supportive feedback and are delighted that you found our proposed sensitivity analysis useful for studying infinite width RNNs.  Below we address each of your comments.
>
> **The proposed method is restricted to the small data setting**: Indeed, a downside shared by all kernel methods is that their computational complexity scales quadratically with the signal length and number of samples. In this first paper on the RNTK, we focused on its derivation, theoretical analysis, interpretation, and proof-of-concept validation experiments. However, we agree that developing a computational framework for the RNTK (and other NTK kernels) to large-scale data is an interesting direction for future research. We also would like to emphasize that, despite its computational limitations, the RNTK can handle data of varying lengths, as highlighted in Theorem 1 and the Remark on page 4, in sharp contrast with other kernels, including the NTK.
>
> **It seems that the proposed RNTK cannot outperform other state-of-the-art (SOA) methods**: We respectfully disagree.  Table 1 demonstrates that, for a wide range of datasets, the RNTK outperforms SOA methods.  We report in Table 1 the aggregate statistics, i.e., the average accuracies, the Friedman ranking, and P90/50 across all datasets, which are common metrics used in the literature to show the superiority of a classifier compared to other classifiers on a set of datasets. The much larger Table 2 in the Appendix reports on the per-dataset accuracies to provide transparency in the results and simpler model comparison on a per-dataset basis.
>
> **Why are GRU and identity RNNs chosen?** The main goal of our paper is to propose a novel kernel based on the RNN and to compare its performance against standard kernels and other NTKs. That being said, comparing RNTK performances with the finite width regime RNNs with Gaussian initialization is also important to see how finite versus infinite width plays into the performances of generalization of architectures, for both theoretical and practical purposes. However, as finite width RNNs can suffer from training instabilities such as vanishing or exploding gradients, we see in Table 1 that Gaussian RNN has the worst performance among all other methods. We added improved versions of recurrent architectures that have more stable training and better generalization ability, such as Identity RNNs and GRU, to see these versions perform in comparison to kernels methods, and more importantly RNTK, mainly to hint practitioners on which method to use on such small size time-series datasets.
>
> **Clarification in Table 2**: Thanks for pointing out the difficulty of interpreting Table 2. We have highlighted the best performer for each dataset in the revised version.

---

### Official Review · AnonReviewer3 · 2020-10-27
**Limited in scope, but interesting, and excellent in the presentation**

**Rating:** 7
**Confidence:** 4

**Review:**

This paper studies the NTK of RNNs in the infinite-width limit, and shows a number of interesting features of such networks, that are somehow surprising knowing the problems of exploding gradients, or knowing the need for independence of parameters in proofs involving the NTK (for the tied weights giving the same as the untied case). While the techniques are not particularly new (they are relying on the applciation of techniques appearing in earlier papers), and the idea to look at RNNs is fairly straightforward, I think this is an interesting and useful paper, with nontrivial estimates (that look correct, although I may need a bit more time to check). The quality of the writing is extremely high, the notations are optimal, and it is overall a pleasure to read, and this paper will serve as an excellent basis for future investigations.

---

> ### Author Response · Authors · 2020-11-15
> **Thank you**
>
> We thank the reviewers for their positive comments about our paper. We are pleased that you find the paper clear and heartily agree that our work provides a new foundation for not only the study of infinite width RNNs but also avenues to explore improved RNN and NTK architectures.

---

### Official Review · AnonReviewer1 · 2020-10-29
**Theoretical results are valuable,  empirical results are insufficient.**

**Rating:** 6
**Confidence:** 3

**Review:**

In this paper the neural tangent kernel for RNNs is derived. It is emphasized how this results in a proper kernel that can handle samples of different lengths.

The  theoretical derivations are correct to the best of my knowledge and help to complete the picture of deep architectures for which the NTK has been derived. This is a valuable contribution.

My main concern is on the empirical side. Since the sinusoid regression is a toy problem I'll focus on the other two:

- Google stock price regression: Appendix A2 doesn't provide much insight into the experiment setup, but it seems that the objective is to predict the stock price of the next day based on the previous ones. I would assume that a strong baseline for this task will be to simply predict the stock price to be the same as the previous day. This is a trivial predictor that requires a training set size of zero. So if you were to plot its performance on Figs 4c and 4d, which (constant) SNR would it yield? I'd be surprised if it is beaten (and that would be a very interesting result), but also I'd be surprised if it's not matched (seems a simple enough predictor to discover if there is no overfitting). However, those graphs show a big spread between different predictors, which is not intuitive given the previous sentences. Can you pleaser report the SNR of "my" trivial predictor and explain the apparent discrepancy?

- The 53 UCR classification datasets: Why those 53? The database contains 128, and your paper states that you took "data sets with fewer than 1000 training samples and fewer than 1000 time steps". But datasets "Adiac" and "Beef" fulfill those requirements and were not taken. On the other hand, "StarLightCurves" doesn't seem to fulfill them (1024 steps) and yet you took it. Can you clearly state how you chose those 53 datasets out of the 128 available?

---

> ### Author Response · Authors · 2020-11-15
> **More empirical results are added**
>
> We thank the reviewers for their constructive comments. Below are our responses for each point.
>
> **Clarifications and baseline Google stock price regression** : We have included more details on the experimental setup in the main text in the revised version. We thank the reviewer for the suggested baseline predictor for the Google stock price regression; we validated it for both the synthetic and the real world datasets and added the corresponding SNRs (Figures 4 a-d). This baseline strategy indeed provides competitive performance for the stock data, since it is well matched to this data. However, it significantly underperforms the other methods we consider (including the RNTK) for the sinusoid example we considered. We added the full details of this experiment in the revised version of the manuscript.
>
> **How we choose datasets from the UCR corpus**: We definitely did not cherry pick the datasets that we showcase in the paper. Our rationale behind the selection of the 53 datasets we considered was as follows.
> First, we selected the datasets from the UCR corpus with fewer than 1000 samples and 1000 time steps so that we could compare the RNTK to a number of other kernel methods (as we point out below, kernel methods of any kind tend to be computationally expensive, and we had to limit the run time involved in comparing a large number of kernels on a large number of datasets -- see our reply to Reviewer 3 below).
> Second, from 88 remaining datasets, we removed datasets with an atypical number of samples versus number of classes (which consequently contain datasets with very few samples), such as the two datasets you have mentioned: Beef with 6 samples per class, Adiac with 10.54 samples per class, and other datasets like WordSynonyms with 10.68 samples per class. In our revised Table 1 and 2, we report results including these three datasets in the revised version:
>
> Beef - RNTK: 90%, NTK: 73.33%, RBF:83.33%, **Poly: 93.33%**, Gaussian RNN: 26.67%, GRU: 36.67%, identity RNN: 46.67%
> Adiac - RNTK: 76.73%, NTK 71.87%, RBF: 73.40%, Poly: **77.75%**, Gaussian RNN: 51.4%, GRU: 60.61%, identity RNN: 16.88%
> WordSynonyms - RNTK: 57.99%, NTK: 58.46%, RBF: 61.13%, Poly: **62.07%**, Gaussian RNN: 17.71%, GRU: 53.76%, identity RNN: 45.77%
>
> In the submitted paper, we also explored the RNTK’s performance with two slightly longer signals – StarLightCures (1024 times steps) and SemgHandSubjectCh2 (1500 time steps) – and observed that the RNTK provides competitive performance.

---

### Decision · Program_Chairs · 2021-01-07
**Final Decision**

**Decision:**

Accept (Poster)

**Comment:**

Reviewers agreed on the value of theoretical contribution, especially the surprising conclusion that the weight-tied and untied RNTK are identical. The empirical results were updated in response to reviewer's suggestion. I believe this would be of interest to ICLR audience.